# Subthalamic, not striatal, activity correlates with basal ganglia downstream activity in normal and parkinsonian monkeys

Marc Deffains[1,2]*, Liliya Iskhakova[1,2], Shiran Katabi[1], Suzanne N Haber[3], Zvi Israel[4], Hagai Bergman[1]

[1]Department of Medical Neurobiology, Institute of Medical Research Israel-Canada, The Hebrew University-Hadassah Medical School, Jerusalem, Israel; [2]The Edmond and Lily Safra Center for Brain Sciences, The Hebrew University, Jerusalem, Israel; [3]Department of Pharmacology and Physiology, University of Rochester School of Medicine, Rochester, United States; [4]Department of Neurosurgery, Hadassah University Hospital, Jerusalem, Israel

**Abstract** The striatum and the subthalamic nucleus (STN) constitute the input stage of the basal ganglia (BG) network and together innervate BG downstream structures using GABA and glutamate, respectively. Comparison of the neuronal activity in BG input and downstream structures reveals that subthalamic, not striatal, activity fluctuations correlate with modulations in the increase/decrease discharge balance of BG downstream neurons during temporal discounting classical condition task. After induction of parkinsonism with 1-methyl-4-phenyl-1,2,3,6-tetrahydropyridine (MPTP), abnormal low beta (8-15 Hz) spiking and local field potential (LFP) oscillations resonate across the BG network. Nevertheless, LFP beta oscillations entrain spiking activity of STN, striatal cholinergic interneurons and BG downstream structures, but do not entrain spiking activity of striatal projection neurons. Our results highlight the pivotal role of STN divergent projections in BG physiology and pathophysiology and may explain why STN is such an effective site for invasive treatment of advanced Parkinson's disease and other BG-related disorders.

*For correspondence: marcd@ekmd.huji.ac.il

**Competing interests:** The authors declare that no competing interests exist.

## Introduction

State-of-the-art basal ganglia (BG) computational models (*Gurney et al., 2015*; *Schultz et al., 1997*) divide the BG network into two functionally related subsystems. First, the main axis (or 'actor' in machine learning terminology) which corresponds to the BG structures that connect state-encoding thalamo-cortical areas to cortical and brainstem motor centers. Second, the neuromodulators (machine learning's 'critics', e.g., the midbrain dopaminergic neurons and striatal cholinergic interneurons) that adjust activity along the BG main axis by encoding a prediction error signal capable of modulating the efficacy of cortico-striatal transmission (*Deffains and Bergman, 2015*; *Reynolds et al., 2001*; *Shen et al., 2008*).

The input structures of the BG main axis (the striatum and subthalamic nucleus, STN) receive considerable glutamatergic inputs from the cortex and the thalamus. The striatum and STN provide major inhibitory GABAergic and excitatory glutamatergic drive respectively to the external segment of the globus pallidus (GPe) and the BG output structures (internal segment of the globus pallidus and substantia nigra reticulata, GPi/SNr) (*Parent and Hazrati, 1995a*, *1995b*). In return, the GPe emits feedback GABAergic projections to the STN (*Carpenter et al., 1981*) and the striatum

**eLife digest** The symptoms of Parkinson's disease include tremor and slow movement, as well as loss of balance, depression and problems with sleep and memory. The death of neurons in a region of the brain called the substantia nigra pars compacta is one of the major hallmarks of Parkinson's disease. These neurons produce a chemical called dopamine, and their death reduces dopamine levels in another area of the brain called the striatum. This structure is one of five brain regions known collectively as the basal ganglia, which form a circuit that helps to control movement.

The most effective treatment currently available for advanced Parkinson's disease entails lowering electrodes deep into the brain in order to shut down the activity of part of the basal ganglia. However, the target is not the striatum; instead it is a structure called the subthalamic nucleus. The striatum and the subthalamic nucleus are the two input regions of the basal ganglia: each sends signals to the other three structures downstream. So why does targeting the subthalamic nucleus, but not the striatum, reduce the symptoms of Parkinson's disease?

To shed some light on this issue, Deffains et al. recorded the activity of neurons in the basal ganglia before and after injecting two monkeys with a drug called MPTP. Related to heroin, MPTP produces symptoms in animals that resemble those of Parkinson's disease. Before the injections, spontaneous fluctuations in the activity of the subthalamic nucleus produced matching changes in the activity of the three downstream basal ganglia structures. Fluctuations in the activity of the striatum, by contrast, had no such effect. Moreover, injecting the monkeys with MPTP caused the basal ganglia to fire in an abnormal highly synchronized rhythm, similar to that seen in Parkinson's disease. Crucially, the subthalamic nucleus contributed to this abnormal rhythm, whereas the striatum did not.

The results presented by Deffains et al. provide a concrete explanation for why inactivating the subthalamic nucleus, but not the striatum, reduces the symptoms of Parkinson's disease. Further research is now needed to explore how the striatum controls the activity of downstream regions of the basal ganglia, both in healthy people and in those with Parkinson's disease.

(*Hegeman et al., 2016*; *Mallet et al., 2012*) as well as massive feedforward GABAergic projections to the GPi/SNr (*Parent and Hazrati, 1995b*). Thus, aside from the action of the BG neuromodulators and lateral connectivity, the increase-decrease balance of spiking activity (I/D balance) of pallidal and nigral neurons is fined-tuned by the inhibitory and excitatory drives exerted by the striatum and STN, respectively. However, how these antagonistic drives operate to convey relevant information from the state-encoding thalamo-cortical areas through the central (GPe) and output (GPi and SNr) BG structures to brain motor centers is still unknown.

Many human disorders are caused by malfunctions of the BG neuromodulators which impact neuronal activity along the BG main axis. Traditionally, in Parkinson's disease (PD), it is assumed that degeneration of midbrain dopaminergic neurons leads to striatal dopamine depletion which provokes a cascade of physiological disturbances in the BG main axis, notably the emergence of synchronized oscillatory activity in the BG and cortical networks (*Levy et al., 2002*; *Nini et al., 1995*; *Oswal et al., 2013*). These abnormal oscillations likely compromise information flow through the BG main axis and result in the release of abnormal commands by BG output structures.

Despite evidence of subthalamic dopamine depletion in PD and its role in the pathophysiology of the disease (*Francois et al., 2000*; *Galvan et al., 2014*; *Rommelfanger and Wichmann, 2010*), the striatum remains the main site of dopamine depletion in human patients and animal models of PD. In addition, the striatum is much larger than the STN ($10^7$ vs. $10^5$ neurons in non-human primates, respectively, *Hardman et al., 2002*). Nevertheless, the STN, not the striatum, is the prime target for deep brain stimulation (DBS) of human patients with advanced PD (*Limousin et al., 1998*; *Odekerken et al., 2016*). Moreover, it has been shown that STN-DBS abolishes abnormal synchronized oscillations in the BG network of animal models of PD (*Meissner et al., 2005*) and human PD patients (*Kühn et al., 2008*; *Wingeier et al., 2006*). These findings suggest that STN plays a pivotal role in the release of commands by BG output structures, but the respective influence of the striatum and STN activity on the activity of the BG central and output structures in PD are still unknown.

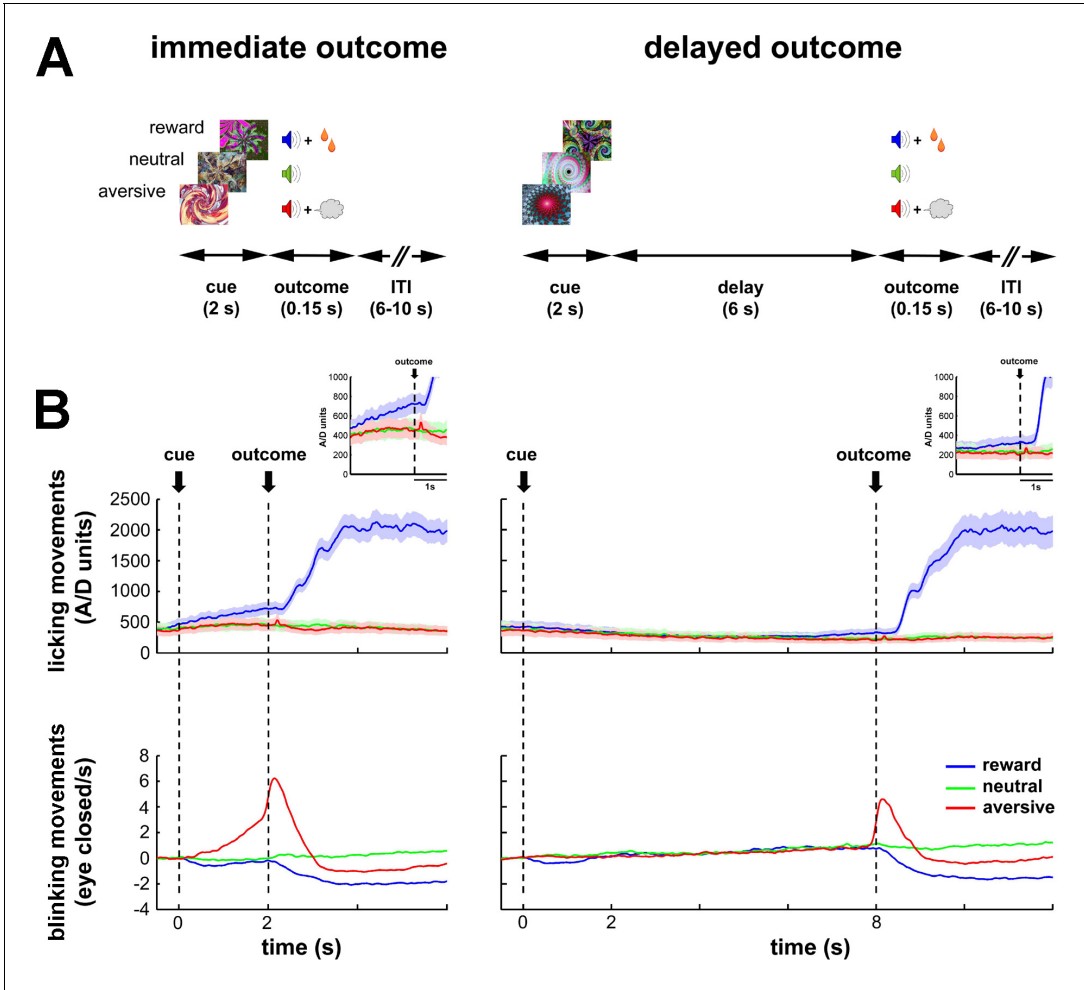

**Figure 1.** Task design and behavioral monitoring. (A) Temporal discounting classical conditioning task. Each trial started with the presentation of a visual cue (2 s) that predicted the delivery of food (reward/appetitive trials), airpuff (aversive trials) or sound only (neutral trials). Cue offset was immediately followed by the outcome period (immediate outcome condition) or by a 6-s delay period which preceded the outcome period (delayed outcome condition). The outcome period (0.15 s) was followed by a variable inter-trial interval (ITI) of 6–10 s. Trial order and ITI duration were randomized. (B) Animals' task performance. Frequency of licking (top) and blinking (bottom) movements over time are aligned to cue onset (time = 0). Time 2 and 8 s correspond to outcome delivery in the immediate and delayed conditions, respectively. Data were averaged for each session (hundreds of trials) and then across sessions (N = 41 and 30, monkey K and S, respectively). Data were grouped since no significant differences were found between the two monkeys. Solid line and shaded envelope represent mean ± standard error of the mean (SEM), respectively. Color code indicates the cue/outcome value: blue-appetitive, green-neutral and red-aversive.

To tackle these issues, we recorded the neuronal activity of the BG input and downstream (central and output) structures of two monkeys engaged in a classical temporal discounting conditioning task (i.e., normal/healthy state, *Figure 1A*). In the task, we manipulated the value of 2-s cues (predicting future appetitive, aversive or neutral outcomes) and the delivery time of the outcome (immediate or 6-s delayed). Then, once we completed the recordings in the normal state, we proceeded to record neuronal activity in the BG network of the same two monkeys after systemic induction of PD symptoms (i.e., parkinsonian state) with 1-methyl-4-phenyl-1,2,3,6-tetrahydropyridine (MPTP). These multi-site recordings in both the normal and parkinsonian states served us to reveal how BG activity propagates along the BG main axis in health and parkinsonism. Moreover, it sheds light on which BG input structure (striatum or STN) is more influential in shaping the activity of the BG downstream structures in the recorded conditions.

## Results

### Neuronal database

After an intensive training period in the task (*Figure 1A*), a recording chamber was attached to the animal's skull to allow access through a burr-hole to all BG structures (*Figures 2A and B*). Then, neuronal activity was recorded in the normal/healthy state while the monkeys were engaged in the task (*Figures 2C and D*) and in the parkinsonian state (Figures 8, 9 and 10A).

In the striatum, we targeted the phasically active medium spiny neurons (MSNs) that correspond to the striatal projection neurons and the tonically active neurons (TANs, presumably the striatal cholinergic interneurons, but see *Beatty et al., 2012*). The striatum is composed of the posterior putamen (motor area), the caudate nucleus (associative area) and ventral striatum/nucleus accumbens (limbic area) (*Parent and Hazrati, 1995a*). In this study, striatal (MSN and TAN) spiking activity was preferentially collected within the posterior putamen. The remaining striatal recordings were made in the caudate nucleus. As previously reported (*Adler et al., 2013a*; *Graybiel et al., 1994*), we did not find significant differences in the activity of MSNs and TANs between these two striatal sub-regions (data not shown). Therefore, we grouped each striatal cell-type recorded from the putamen and caudate nucleus for further analysis.

We also recorded the activity of the neurons of the STN and the high-frequency discharge (HFD) neurons of the GPe, GPi and SNr. HFD neurons of the GPe, GPi and SNr comprise >85% of the neurons in these structures and, like striatal MSNs and STN neurons, are probably the projection neurons of these structures. Thus, a unique, and very advantageous for the current study, feature of the BG network is that each BG nucleus forms a single layer network; in other words, the BG projection neurons are innervated by the projections of the upward structures. Namely, the activity of striatal and STN projection neurons directly affects the activity of the recorded GPe, GPi and SNr projection neurons. We used this property to infer the relative influence of the BG input structures (striatum and STN) on the activity of the central (GPe) and output (GPi and SNr) structures of the BG network.

All recorded neurons were analyzed offline for discharge rate stability and isolation quality (*Joshua et al., 2007*) to guarantee the data quality (see Materials and methods). Neuronal database details are given in *Table 1*.

### Behavioral policy changes between the immediate and delayed conditions of the behavioral task

During the task, we systematically recorded licking and blinking movements to assess the animals' performance (*Figure 1B*). In both conditions (immediate and delayed) of the behavioral task, the frequency of the licking and blinking movements increased in response to food and airpuff delivery, respectively. Remarkably, the animals responded with appropriate anticipatory licking and blinking behavior to the outcome delivery in the immediate condition. In contrast, in the delayed condition, the animals did not display any robust anticipatory licking and blinking movements during the cue and delay periods. Together, these behavioral results indicate that the animals learned the cue values (appetitive, aversive or neutral) and differentiated between cues predicting an immediate or a delayed outcome. However, the introduction of a 6-s delay period after cue offset probably compromised the temporal predictability of the outcome delivery in the delayed condition.

### Persistent modulations of activity along the BG main axis

*Figure 3* depicts the mean (for each BG neuronal assembly) relative (gray) and absolute (green) peri-stimulus histograms (PSTHs). The mean relative PSTH is calculated as the arithmetic average of the single neurons' PSTHs (*Figure 2D*) and is more standard in neuroscience. The mean relative PSTH reflects the common assumption that the on-going activity of the studied population is close to zero and that the studied neurons excite their neuronal targets that act as integrate-and-fire neurons (*Izhikevich, 2007*). On the other hand, most BG neurons fire at tonic high frequencies (*Figures 2C and D*) and a large fraction of their responses to behavioral events consists of decreases in discharge rate (*Adler et al., 2012*; *Espinosa-Parrilla et al., 2013*; *Joshua et al., 2009b*; *Turner and Anderson, 2005*). Thus, neurons of the BG network might follow resonance rather than integrate-and-fire rules when they are activated by their afferents (*Izhikevich, 2007*). We therefore calculated the mean absolute PSTH (absolute deviation from the baseline of the firing rate, see Materials and methods)

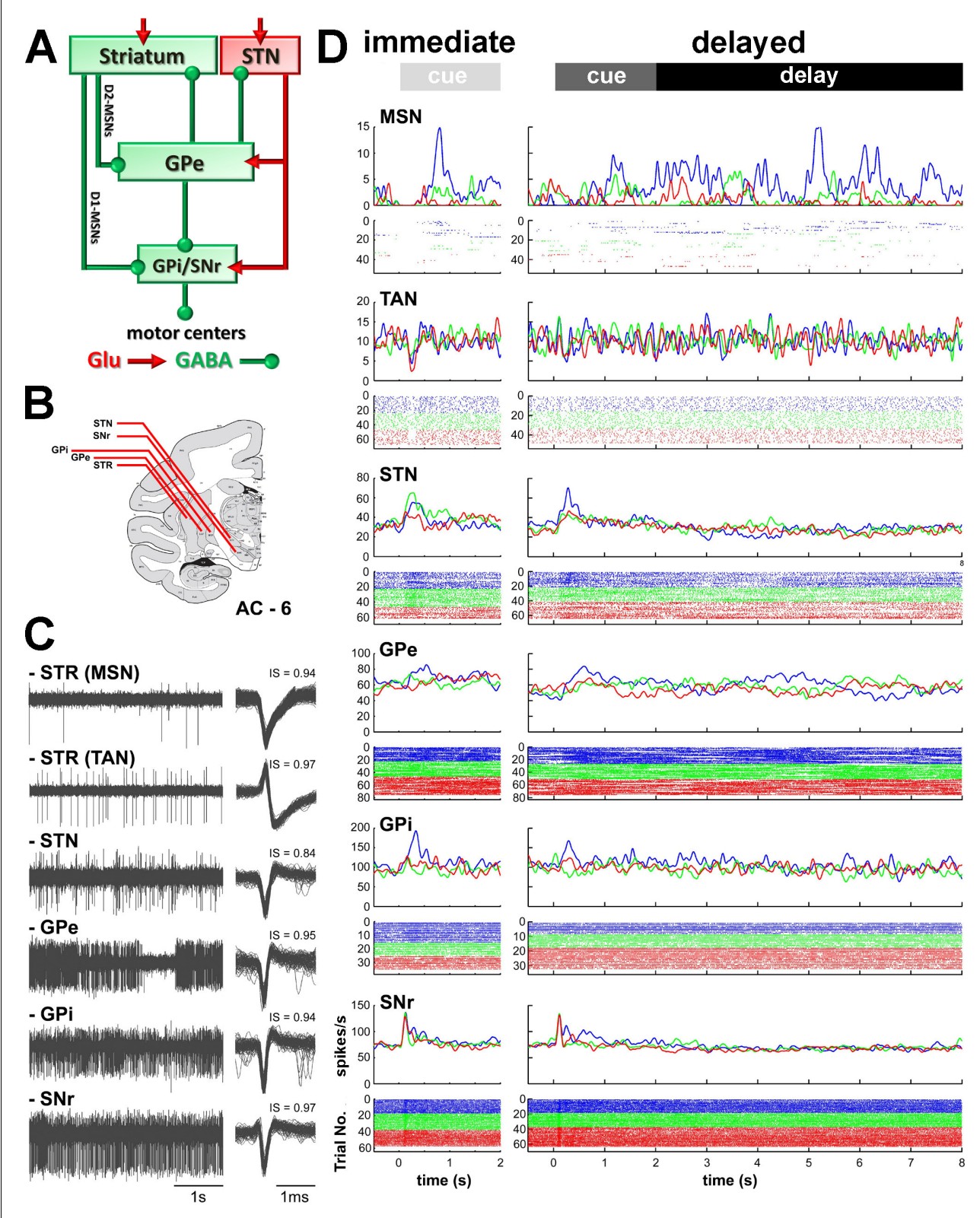

**Figure 2.** Recordings sites and neuronal activities in the BG network. (A) Schematic model of BG functional connectivity. Glutamatergic and GABAergic connections are shown in red and green, respectively. D1-/D2-MSNs: striatal medium spiny (projection) neurons expressing D1/D2 dopamine receptors; STR: striatum; STN: subthalamic nucleus; GPe-i: external or internal segment of the globus pallidus; SNr: substantia nigra reticulata. (B) Representative coronal section - 6 mm from the anterior commissure (AC - 6), adapted from *Martin and Bowden (2000)*. (C) Left, examples of 4 s of multi-unit activity,

*Figure 2 continued on next page*

*Figure 2 continued*

online filtered between 250 and 6000 Hz. Right, 100 randomly chosen superimposed waveforms of the extracellular action potentials of a cell sorted from its multi-unit activity. IS indicates the isolation score of the sorted cell. (D) Peri-stimulus time histograms (PSTHs) and raster plots of 6 cells (single units) recorded in the different structures of BG, aligned to cue onset (time = 0). Time 2 and 8 s indicate outcome delivery in the immediate and delayed outcome conditions, respectively. PSTHs were built by summing activity across trials at a 1-ms resolution and then smoothing with a Gaussian window (SD = 20 ms). Each line of dots in the raster plots corresponds to activity during a trial and each dot represents a single spike. Color code indicates trial type according to outcome: appetitive trials in blue, neutral trials in green and aversive trials in red.

of the BG neurons recorded in our study. These absolute PSTHs revealed that the average population response to the cue of each neuronal assembly of the BG main axis persisted until the outcome delivery in both conditions (immediate and delayed) (*Figure 3*, green lines). Therefore, these persistent modulations of activity along the BG main axis temporally bridged sensory visual information (provided by cue presentation) and the actions (licking and blinking movements) wired to outcome delivery and continued even when the visual information was no longer available during the 6-s delay period in the delayed condition. The persistent responses in the BG main axis contrast with the transient response of the TANs (i.e., one of the BG neuromodulators) to the cue onset (*Figure 3*).

For each neuronal assembly of the BG main axis, the absolute response of the population was systematically larger than the relative response (*Figure 3*), thus suggesting in line with previous studies (*Adler et al., 2012*; *Espinosa-Parrilla et al., 2013*; *Joshua et al., 2009b*; *Turner and Anderson, 2005*) that single neuronal responses in the BG main axis consist of either increases or decreases in the firing rate. We found that neurons of the BG main axis exhibited diverse patterns in timing and polarity (increase/decrease) of discharge modulations rather than coordinated sustained modulations of the discharge rate (*Figure 4*).

In both conditions, we also found that the fraction of neurons that modulated their activity (regardless of the polarity of the modulation) at each time bin (20 ms), from cue onset to outcome delivery, systematically exceeded chance level (i.e., $p < 0.05$, the values lay beyond two standard deviations of the mean, empirical 68-95-99.7 rule) for all neuronal assemblies of the BG main axis (*Figure 5*, left and center histograms). Except for striatal MSNs, analysis of variance (ANOVA) revealed a significant effect of task period (cue in the immediate condition and cue and delay in the delayed condition) on the fraction of responsive bins (one-way ANOVA, $p < 0.001$) for all other neuronal assemblies of the BG main axis. The fractions of responsive bins in the STN and downstream structures decreased during the delay period ($p < 0.001$, *post hoc* comparisons, Bonferroni corrected, *Figure 5*, right bar graphs). Thus, while the fraction of modulated MSNs remained unchanged over time, STN, GPe, GPi and SNr neurons exhibited comparable changes following cue offset in the delayed condition.

**Table 1.** Neuronal database. N is the number of recorded neurons that passed inclusion criteria. The recording span represents the stable recording period (in seconds) for each neuron. The isolation score ranges from 0 to 1. The recording span and isolation score were averaged for each structure and state. Values are means ± standard deviation (SD). For each neuronal assembly, statistics were calculated and are presented for both monkeys.

| Neuronal assembly | before MPTP | | | after MPTP | | |
|---|---|---|---|---|---|---|
| | N | Recording span (s) | Isolation score | N | Recording span (s) | Isolation score |
| striatum (MSN) | 150 | 1096.8 ± 564.8 | 0.85 ± 0.10 | 128 | 950.6 ± 616.1 | 0.85 ± 0.12 |
| striatum (TAN) | 116 | 1568.8 ± 907.6 | 0.89 ± 0.08 | 81 | 1217.8 ± 708.8 | 0.89 ± 0.09 |
| STN | 103 | 1080.0 ± 411.1 | 0.77 ± 0.12 | 111 | 780.0 ± 317.7 | 0.80 ± 0.12 |
| GPe | 182 | 1760.4 ± 978.2 | 0.91 ± 0.08 | 105 | 877.7 ± 332.7 | 0.91 ± 0.09 |
| GPi | 119 | 1437.0 ± 607.9 | 0.89 ± 0.08 | none | | |
| SNr | 110 | 1205.2 ± 366.4 | 0.90 ± 0.08 | 121 | 874.7 ± 445.4 | 0.90 ± 0.09 |

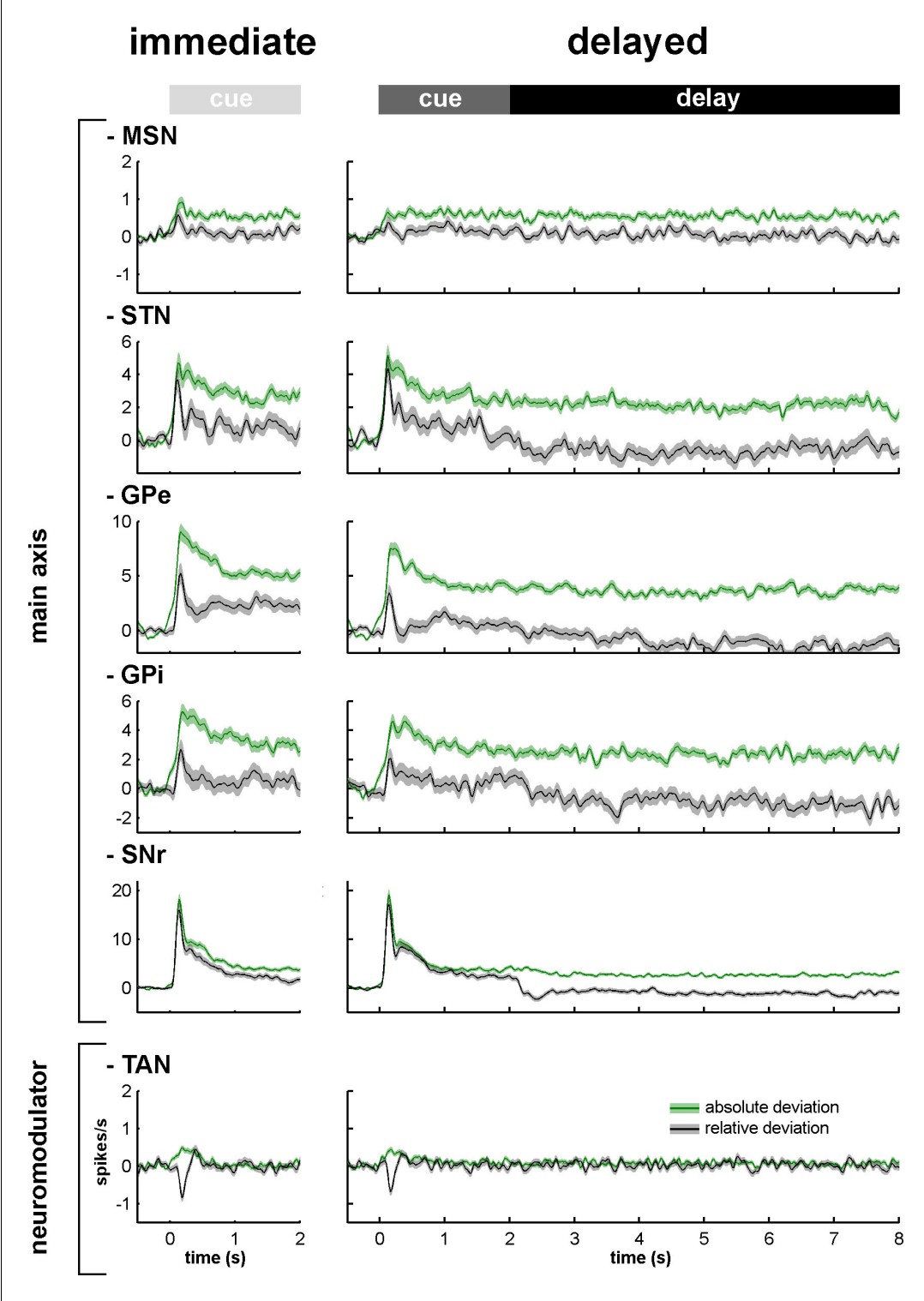

**Figure 3.** Relative and absolute population responses to the task behavioral events. The response of a single neuron to the cue was calculated as the relative or absolute deviation from the baseline of the firing rate. The baseline firing rate was calculated during the 500 ms prior to cue onset. For each condition (immediate/delayed), each neuron was tested three times (appetitive, neutral and aversive trials). Relative (black) and absolute (green) population responses to the cue were defined as the average of the relative or absolute responses of all neurons of the same structure, respectively. The shaded areas mark SEMs.

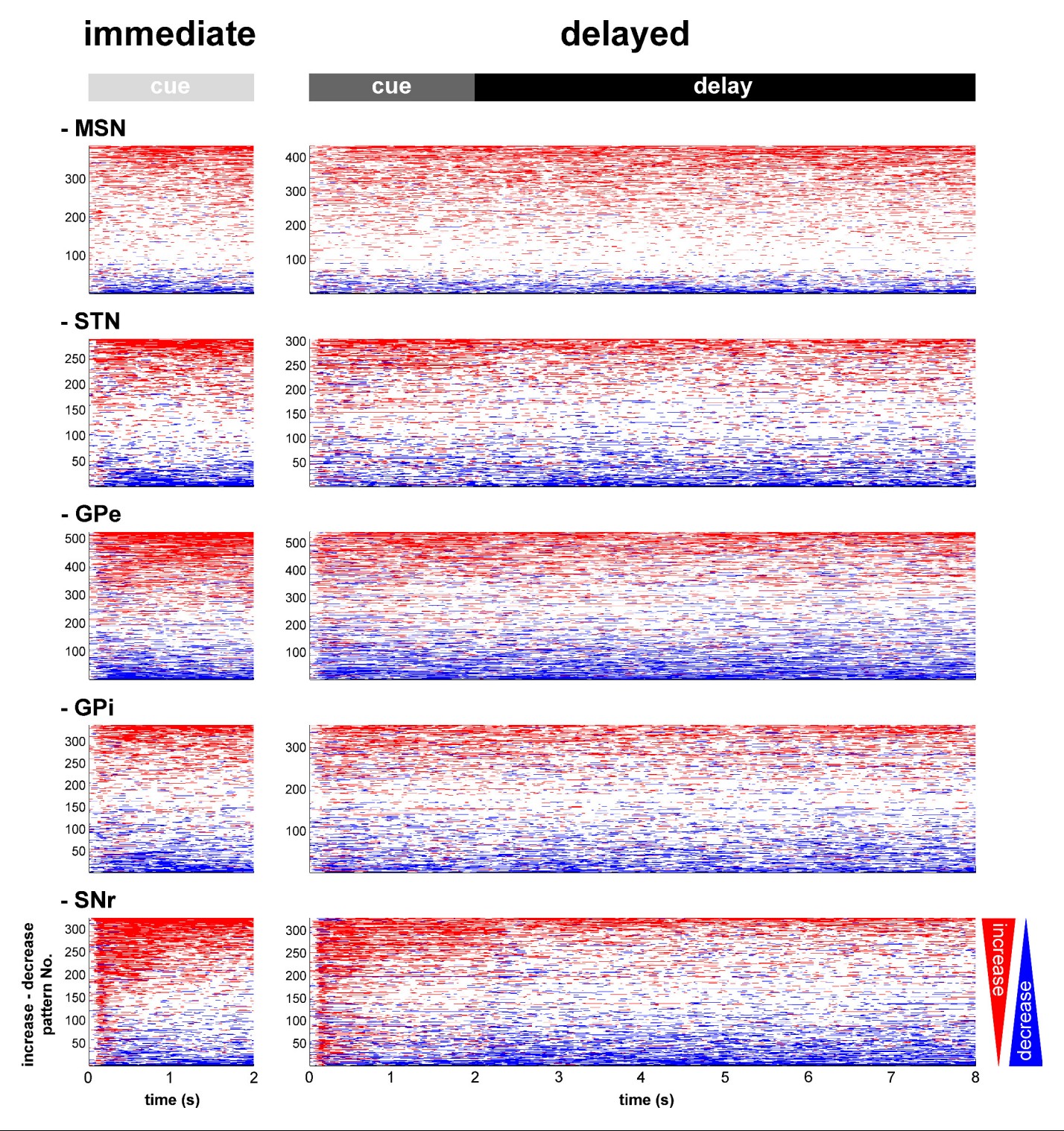

**Figure 4.** Temporal patterns of increases and decreases in activity of the BG neuronal assemblies. In the (red/blue) surface plot, each row represents the time course of increases (red) and decreases (blue) in activity of a single neuron from cue onset to outcome delivery. Only modulated neurons are displayed. A neuron was considered modulated if at least one responsive bin was detected past the cue onset (for each neuronal assembly, >80% of the neurons were modulated whatever the cue value, in both conditions). The temporal patterns of increases and decreases in activity are ordered such that modulatory activities consisting of longer increases are at the top and ones consisting of longer decreases are at the bottom.

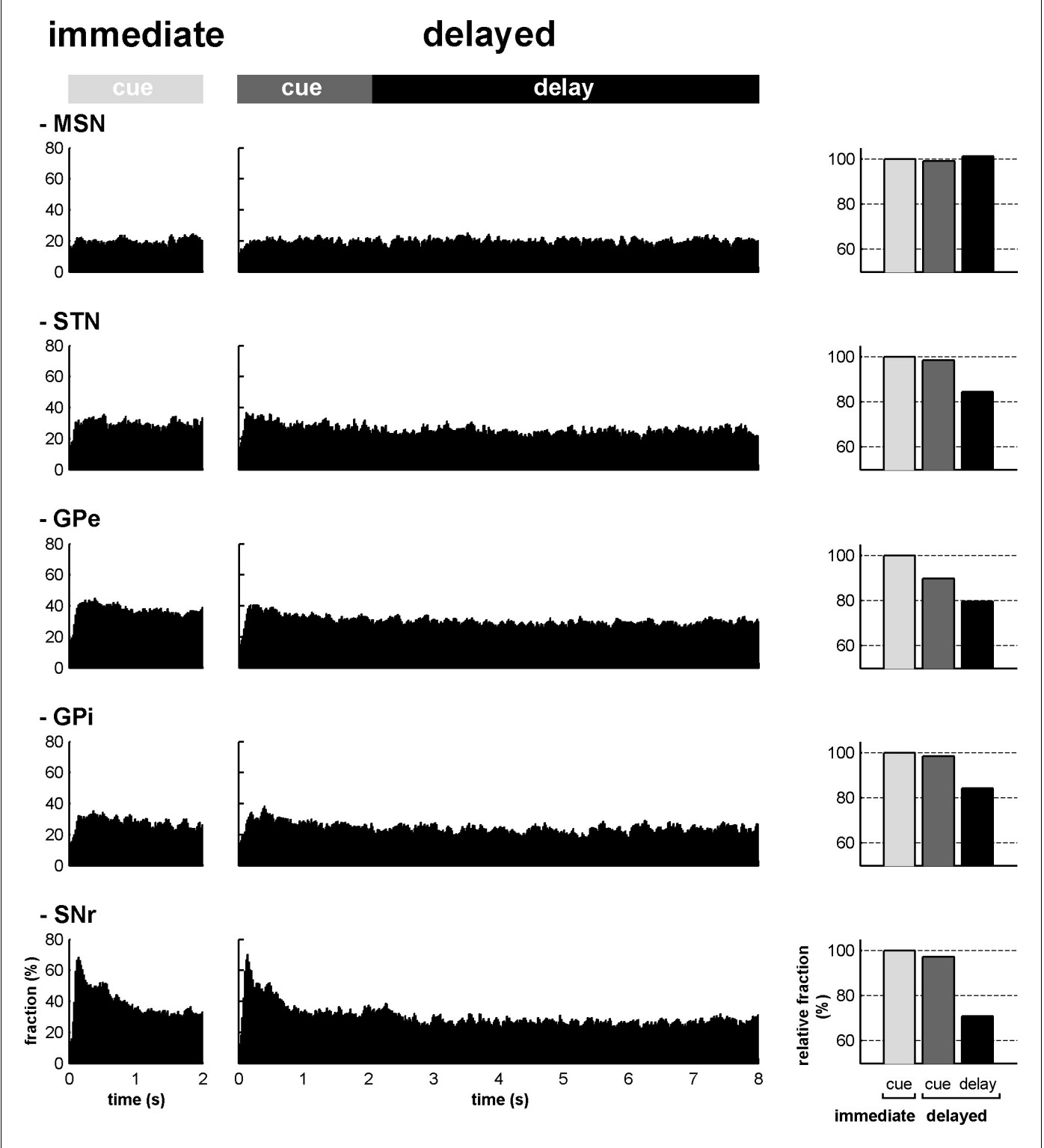

**Figure 5.** The BG main axis exhibits persistent modulations of activity. For each neuronal assembly of the BG main axis, the black histograms (left) represent the fraction of responsive neurons (in the appetitive, neutral and aversive trials) at each time bin (20 ms) from cue onset to outcome delivery in both conditions. The bar plot (right) depicts the mean fraction of responsive neurons (average of the fractions of responsive neurons over a task period) in the different task periods (cue period of the immediate condition and cue and delay periods of the delayed condition) compared to the mean fraction of responsive neurons in the cue period of the immediate condition.

*Figure 5 continued*

The following source data is available for figure 5:

**Source data 1.** Fraction of responsive neurons during the different task periods.

## Time-varying increase-decrease balance of spiking activity along the BG main axis

Theoretical (*van Vreeswijk and Sompolinsky, 1996*) and intracellular recording (*Taub et al., 2013*) studies usually quantify changes in membrane potential as an excitation/inhibition balance (E/I balance). However, using extracellular recording method, we can only record changes in the discharge rate of neurons but not the corresponding changes in synaptic and membrane potentials. Since both an increase in excitation and an abolition of inhibition can lead to an increase in discharge rate (and vice versa for decreases in discharge rate), we used the more conservative nomenclature of the increase/decrease discharge rate balance (I/D balance).

To assess the time-varying I/D balance within the different neuronal assemblies of the BG main axis, we calculated the fraction (*Figure 6—figure supplement 1*) and the magnitude (*Figure 6—figure supplement 2*) of increases and decreases in spiking activity. This was done for each neuronal assembly, from cue onset to outcome delivery, in the two behavioral conditions (immediate and delayed). This way, we examined the I/D balance (i.e., relative increase vs. decrease dominance of activity as measured by the fraction weighted by the magnitude) over time along the BG main axis (*Figure 6*, left and central histograms). We found that during the cue periods in both conditions, increases in activity (positive I/D balance) strongly predominated in all BG structures (*Figure 6*, right bar plots). In contrast, during the delay period, although the I/D balance was still strikingly biased towards increases in activity for the MSNs, this imbalance was significantly reduced in the STN and reversed in the GPe, GPi and SNr (one-way ANOVA, $p<0.001$, with Bonferroni correction for multiple comparisons, *Figure 6*, right bar plots). Thus, whereas the inhibitory GABAergic drive from the striatum was unchanged over the different task periods, the STN excitatory glutamatergic drive was reduced when moving from the cue to the delay periods. This reduced STN excitatory drive was concurrent with a reduction in the activity in the BG downstream structures (*Figures 3*, *5* and *6*) and with the animals' modified behavioral policy (*Figure 1B*).

## Similarity/dissimilarity of the neuronal responses between BG input and downstream structures

In the previous paragraph, we have analyzed the I/D balance of the different populations of BG neurons and concluded that the I/D balance of STN neurons, not MSNs, resembled the I/D balance of the BG downstream neurons. However, we have not used formal similarity/dissimilarity test (e.g., cross-correlation analysis). Here, the cross correlation function was calculated between the neuronal responses (binned relative PSTHs) of (non-simultaneously recorded) BG input-downstream neuron pairs. For this purpose, the BG neuronal responses were categorized as a function of their principal polarity as either increases or decreases (see Materials and methods), and the cross-correlation (similarity/dissimilarity) was tested between neuronal responses with similar and opposite principal polarities.

We found that increases dominated during the cue periods in both conditions ($\chi^2$ test, $p<0.05$, for each BG neuronal population, *Figure 7A*). During the delay period, except for striatal MSNs where increases remained preponderant ($\chi^2$ test, $p<0.001$, *Figure 7A*), this dominance of the increases disappeared for STN, GPe and GPi neurons and decreases even became dominant for SNr neurons ($\chi^2$ test, $p<0.001$, *Figure 7A*).

The similarity coefficients (i.e., the zero lag coefficients of the cross correlation functions) of the neuronal responses of the BG input-downstream neuron pairs exhibiting similar or opposite polarity are shown in *Figure 7B*. Remarkably, for each task period, two-way ANOVA revealed a significant and systematic effect of the BG input (striatum vs. STN) on the similarity coefficient of the neuronal responses of the BG input-downstream neuron pairs (two-way ANOVA, $p<0.001$). Nevertheless, this significant BG input effect varied depending on the polarity of the neuronal response of the BG input neurons that composed the BG input-downstream pairs. For each task period, we found that

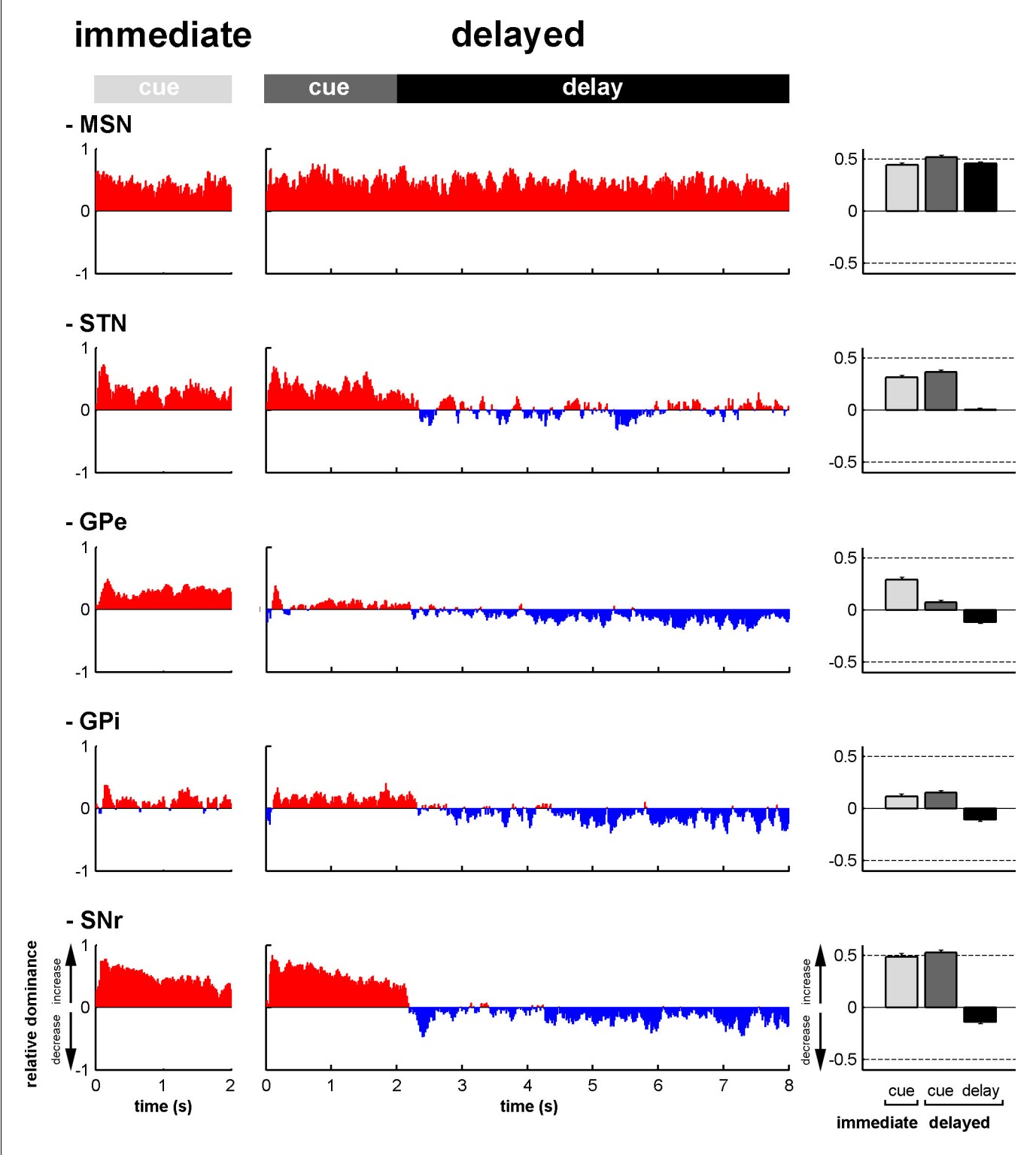

**Figure 6.** Modifications in subthalamic activity are concomitant with reversal of the increase/decrease balance of activity in the BG downstream structures. Left, relative dominance of the increases and decreases in spiking activity (I/D balance) from cue onset (time = 0) to outcome delivery (time = 2 or 8 s in the immediate and delayed conditions, respectively), along the BG main axis. On the y-axis, as values approach 1, increases prevail over decreases and vice-versa as values approach -1. Values close to 0 indicate equal weight of increases and decreases. Right, the mean value of the I/D balance for the different task periods in each neuronal assembly. Error bars represent SEMs.

*Figure 6 continued on next page*

*Figure 6 continued*

The following source data and figure supplements are available for figure 6:

**Source data 1.** Increase/decrease balance of activity during the different task periods.

**Figure supplement 1.** Fraction of increases and decreases in activity along the BG main axis.

**Figure supplement 2.** Magnitude of increases and decreases in activity along the BG main axis.

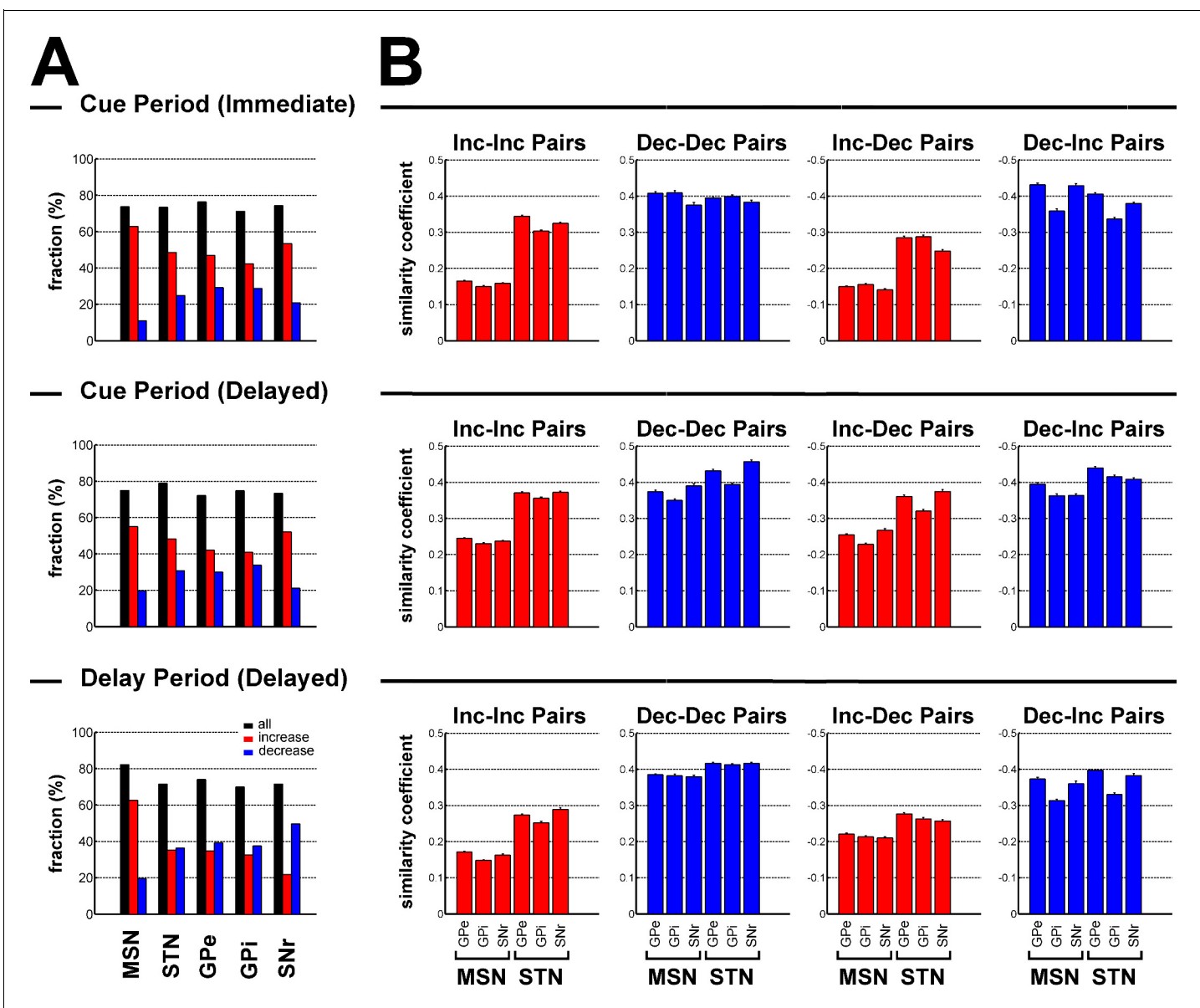

**Figure 7.** Subthalamic, not striatal, activity fluctuations correlate with modulations in BG downstream activity. (**A**) Fraction of neuronal responses depending on their principal polarity along the BG main axis for the different task periods. The sum of the fractions of increases (red) and decreases (blue) is shown in black. (**B**) Mean similarity coefficient of the neuronal responses of striatal (MSN) or subthalamic - BG downstream neuron pairs exhibiting similar (Increase-Increase or Decrease-Decrease pairs) and opposite (Increase-Decrease or Decrease-Increase pairs) polarity, for the different task periods. The similarity coefficient ranges from −1 (i.e., opposite neuronal activity) to +1 (i.e. same neuronal activity).

subthalamic, rather than striatal (MSN), increases correlated with BG downstream increases (Inc-Inc pairs) and decreases (Inc-Dec pairs), while striatal and subthalamic decreases (although there was a significant BG input effect) correlated almost to an equal extent with BG downstream neuronal responses, regardless of their principal polarity (both Dec-Dec and Dec-Inc pairs). However, given the relatively low fraction of striatal decreases (*Figure 7A*), we concluded that subthalamic, but not striatal MSN, neuronal responses (increases or decreases) were more strongly correlated to the BG downstream neuronal responses, regardless of their polarity.

## Spiking oscillatory activity within the BG network in parkinsonism

We used systemic MPTP injections (see Materials and methods) to induce a severe parkinsonian state. F-18 FDOPA positron emission tomography (Monkey K, *Figure 8A*, right) and post-mortem *tyrosine hydroxylase immunohistochemistry* (Monkey S, *Figure 8B*, right) revealed large dopamine depletion in the dorsal striatum (i.e., caudate nucleus and putamen) following the MPTP treatment. Our animals developed all major motor signs of PD, including bradykinesia/akinesia, rigidity, abnormal flexed posture and low-frequency tremor (*Figure 9*). In the parkinsonian state, given the severity of the motor symptoms, the animals were unable to execute the behavioral task and all recordings were made in a quiet alert state. After the initiation of dopamine-replacement therapy, neuronal recordings were made off dopaminergic medication (see Materials and methods) in order to mimic the recording conditions in human parkinsonian patients undergoing DBS surgery. Data collected before and after initiation of dopamine-replacement therapy (in the OFF state) were grouped since no significant difference was detected between these two parkinsonian conditions.

Theoretical studies have shown that neural oscillations can emerge at the population level in networks of neurons exhibiting an irregular (i.e., non-oscillatory) discharge pattern and a low firing rate (*Brunel and Hakim, 2008*; *Kopell and LeMasson, 1994*). In addition, it has been reported that multi-unit cross-correlations might be a more sensitive detector of neuronal relationships than single-unit cross-correlations (*Bedenbaugh and Gerstein, 1997*; *Gerstein, 2000*). Indeed, most intra-operative human physiological studies during DBS procedures are conducted at the level of multi-unit activity (MUA). For these reasons and given the very low firing rate of the MSNs, we investigated multi-unit oscillatory activity recorded in the vicinity of the recorded BG cells rather than single-unit oscillatory activity (*Figure 10A*). Using the MUA recording in the vicinity of a well-isolated single unit better guarantees the location of the recording and rules out shifts in electrode position during the recording.

Comparison of the mean power spectral densities (PSDs) of the MUA at the different stages of the BG main axis before and after MPTP intoxication revealed the emergence of 8–15 Hz oscillations in the STN, GPe and SNr (GPi activity was not recorded in the MPTP-treated monkeys), but not in the recordings from the areas surrounding the striatal MSNs (*Figure 10B*). The distributions of the oscillation frequencies were similar for the STN, GPe and SNr, and peaks of the distribution were found at 10 Hz for the three structures (*Figure 10B*).

Despite the absence of significant oscillations in MSN spiking activity (comprising > 95% of the striatal cells) in the dopamine-depleted striatum, we also found 10 Hz oscillations of striatal TANs after MPTP treatment (*Figure 10B*, insets), indicating that abnormal oscillatory activity did not spare the striatum.

Accordingly, we found that the peak values of the PSD of the spiking activity (in the 8–15 Hz range, see Materials and methods) were significantly higher in parkinsonism (after MPTP) than in healthy state (before MPTP) for TANs, STN, GPe and SNr neurons (two-sample t-test, p<0.01, *Figure 11*), but not for MSNs. Thus, although low beta (8–15 Hz) oscillations were found in the spiking activity of the STN, downstream BG structures, and even striatal TANs, MSNs (recorded in striatal areas with TAN oscillations) did not exhibit such abnormal oscillatory spiking activity.

## LFP oscillations and spike-field coherence within the BG network in parkinsonism

After MPTP intoxication, we also found exaggerated 8–15 Hz oscillations of local field potentials (LFPs) recorded in all BG structures, including the striatum in the vicinity of both TANs and MSNs (*Figure 12*). LFPs are commonly attributed to the sub-threshold (synaptic) activity of the recorded structure. Therefore, we further performed a spike-field coherence analysis to determine whether

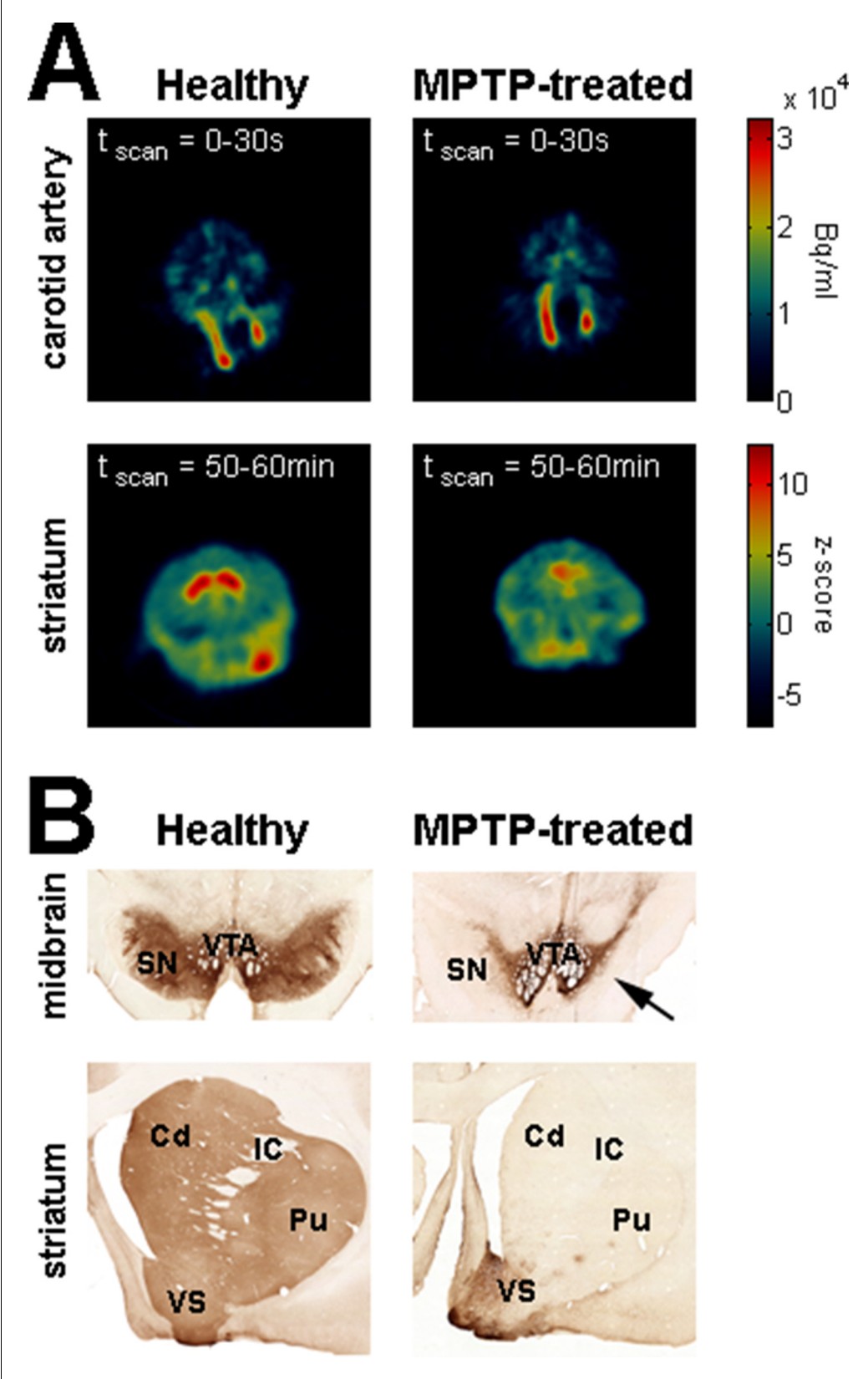

**Figure 8.** Dopamine depletion assessment in parkinsonism. (**A**) Dynamic positron emission tomography (PET) of healthy monkey's brain (left) and MPTP-treated monkey's brain (Monkey K, right). The top panels are PET images showing the regional distribution of F-18 FDOPA at the carotid level
*Figure 8 continued on next page*

*Figure 8 continued*

during the first 30 s following the injection of the radioligand. The color bar (top panels) indicates the radioactivity concentration in Bq/ml. The bottom panels are PET images (acquired from 50 to 60 min after the injection of the radioligand) showing the regional distribution of F-18 FDOPA at the striatal level. Radioactivity concentration was Z-normalized, using the radioactivity concentration in the region of interest (ROI) for the cerebellum (i.e., region of non-specific F-18 FDOPA uptake). The color bar (bottom panels) indicates the relative radioactivity concentration in z-score. (**B**) Photomicrographs of tyrosine hydroxylase (TH) staining demonstrating the loss of the midbrain dopaminergic neurons of the ventral tier of the substantia nigra compacta in the MPTP-treated monkey (Monkey S, right) compared with a healthy animal (left). The top and bottom photomicrographs illustrate the midbrain and (rostral) striatum levels, respectively. After MPTP intoxication, TH-positive cells were lost in the ventral tier of the substantia nigra compacta (see arrow) and relatively spared in the ventral tegmental area. Note the lack of TH-positive staining in the dorsal striatum (caudate nucleus and putamen) compared to the ventral striatum (particularly the shell of nucleus accumbens) in the MPTP-treated monkey. Cd: caudate nucleus; IC: internal capsule; Pu: putamen; SN: substantia nigra; VTA: ventral tegmental area.

the spiking activity of the neurons was synchronized (phase locked) to the LFP recorded in their surrounding areas.

*Figure 13* depicts an example of STN spiking activity after MPTP intoxication that coincides with the negative waves of the 8–15 Hz oscillatory LFP recorded in the vicinity of the neuron. Estimates of the coherence between spike trains of the isolated units (single-unit activity) and LFPs revealed that except for MSNs, peak coherence values in the 8–15 Hz range increased significantly after MPTP

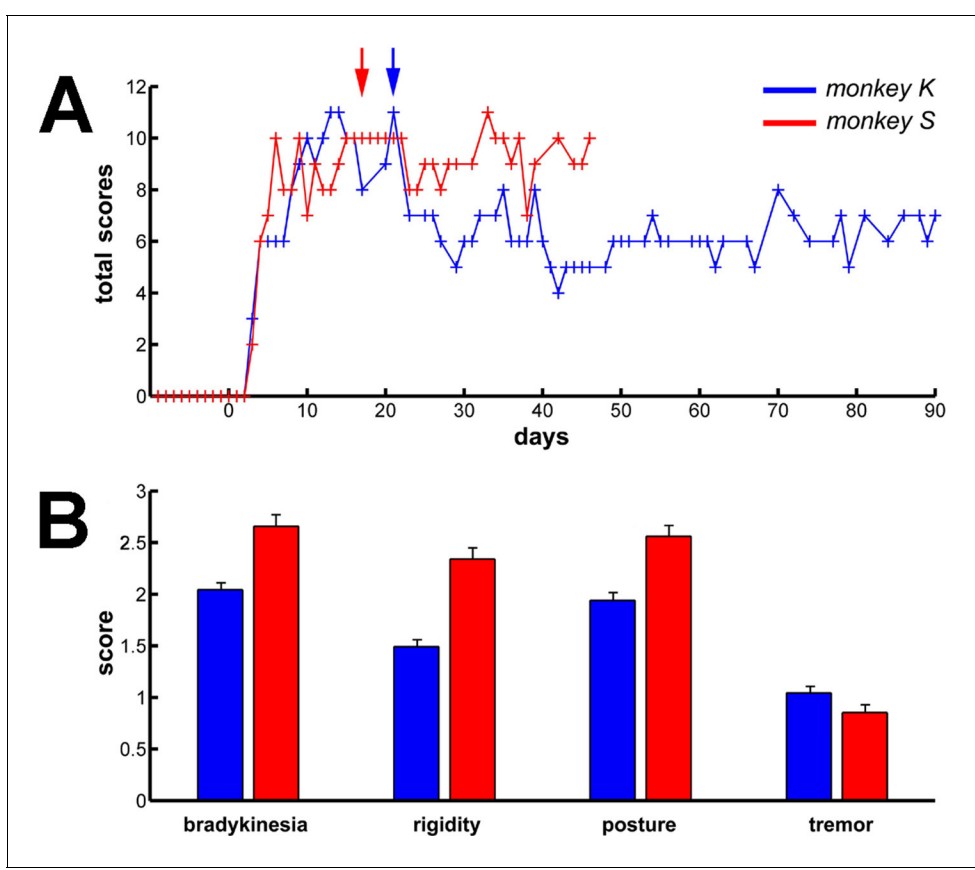

**Figure 9.** Long term and persistent parkinsonian motor symptoms following MPTP intoxication. (**A**) Evolution of the sum of the scores of the major parkinsonian motor symptoms (bradykinesia, rigidity, posture and tremor) in each monkey before and after MPTP intoxication, up to the completion of the recordings. Day 0 indicates the first MPTP injection. Arrows mark the first day of dopamine-replacement therapy for each monkey. After initiation of dopamine-replacement therapy, scoring of the parkinsonian motor symptoms was made off dopaminergic medication (overnight washout >12 hr). The level of parkinsonism was assessed by using a primate parkinsonism scale that rates each motor parkinsonian symptom (bradykinesia, rigidity, posture and tremor) from 0 (normal) to 3 (severe). Hence, the minimum score is 0 and the maximum is 12. (**B**) Mean scores for the four motor symptoms (ranges from 0-normal to 3-severe) during the complete recording period in each MPTP-treated monkey. Error bars represent SEMs.

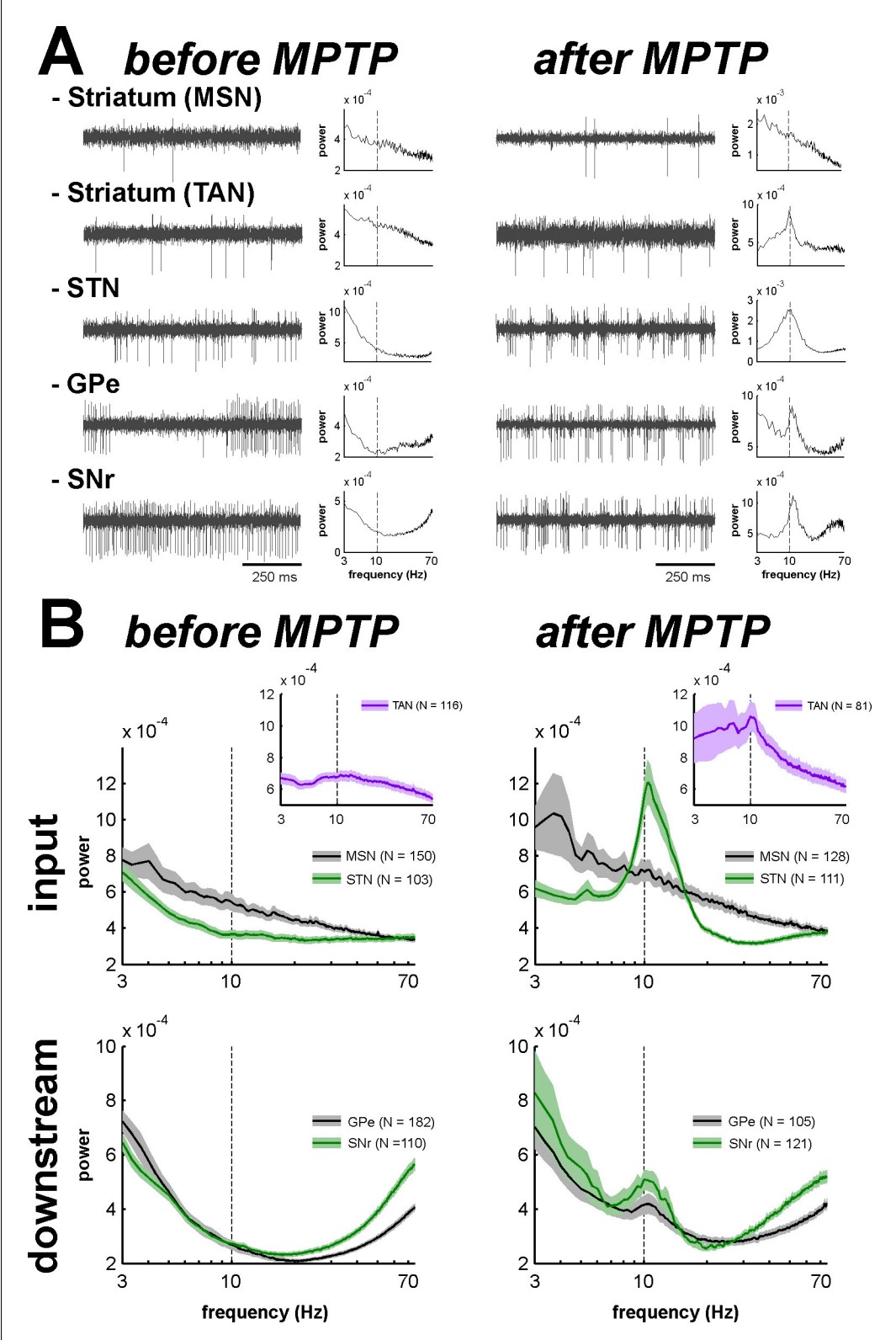

**Figure 10.** Abnormal oscillatory spiking activity is not shared by all BG neuronal components in parkinsonism. (**A**) Examples of 1 s traces of multi-unit activity recorded in the vicinity of an isolated neuron (striatal MSN, striatal TAN, STN, GPe and SNr neurons) in the normal (before MPTP, left) and parkinsonian (after MPTP, right) state and PSDs of the respective multi-unit activity. Multi-unit activity was online filtered between 250 and 6000 Hz. PSD was calculated over the complete recording span of the multi-unit activity constrained by the period of stable discharge rate and satisfactory isolation

*Figure 10 continued on next page*

*Figure 10 continued*

quality of the isolated neuron. Abscissas of the PSDs are in log scale. (B) Average PSDs of activity of MSN, STN (BG input stage, top panels), GPe and SNr (BG downstream stages, bottom panels) recordings, before and after MPTP intoxication (left and right, respectively). Average PSD of activity of striatal TANs is in the inset of the top panels. Abscissas are in log scale. The shaded areas mark SEMs. N is the number of recording sites averaged.

The following source data is available for figure 10:

**Source data 1.** Power spectral densities of the multi-unit activity before and after MPTP.

intoxication (two-sample t-test, p<0.001) for all other BG neuronal assemblies (*Figure 14*, left plots). To guarantee that these spike-field coherence results were not confounded by the slow discharge rate of the striatal MSNs, we also examined the spike-field coherence between the MUA recorded in the striatum (and elsewhere) and the LFP (*Figure 14—figure supplement 1*, left plots). We found that using spike train of the isolated units or MUA for the calculation of the spike-filed coherence yielded the same qualitative results.

After MPTP intoxication, despite the emergence of exaggerated 8–15 Hz LFP oscillations, not all LFPs expressed 8–15 Hz oscillatory activity (*Figure 12*). Therefore, one might suggest that MSN spike-striatal field coherence (*Figure 14*) could materialize only when the striatal LFP (recorded in the vicinity of the MSNs) oscillated. In other words, the current average analysis might not reflect a small population of MSNs that synchronized their activity with the robust 8–15 Hz oscillatory LFPs. To test this hypothesis, we correlated the strength of the LFP oscillations and the degree of spike-field coherence in the striatum (and elsewhere) of the parkinsonian monkeys. We found that regardless of the BG neuronal component (i.e., including striatal MSNs) the degree of synchronization between the single-unit (*Figure 15*) or multi-unit (*Figure 15—figure supplement 1*) spiking activity and the LFP oscillations was not (linearly) related to the strength of the 8–15 Hz LFP oscillations. This rules out the possibility that synchronization between MSN spiking activity and striatal LFP emerged only when LFP recorded in the vicinity of the neuron exhibited robust 8–15 Hz oscillatory activity.

Calculation of the phase of the spiking activity relative to the LFP at the frequency of the coherence peak values in the 8–15 Hz range revealed that LFP relative phase values of the single-unit

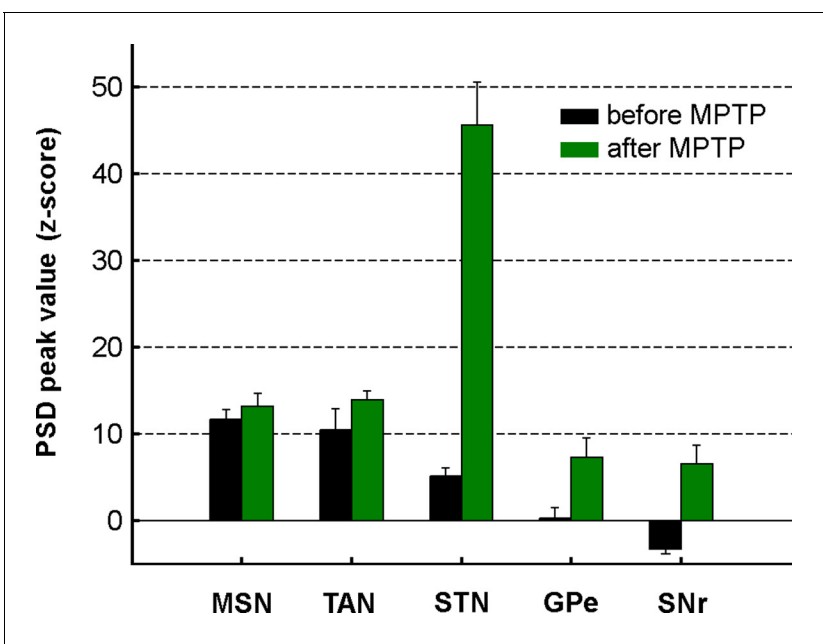

**Figure 11.** Mean PSD peak value in the 8–15 Hz range for the BG neuronal components before and after MPTP intoxication. Mean PSD peak value in the 8–15 Hz range is defined as the average of the PSD peak values in the 8–15 Hz range (Z normalized) of all the PSDs. Error bars represent SEMs.

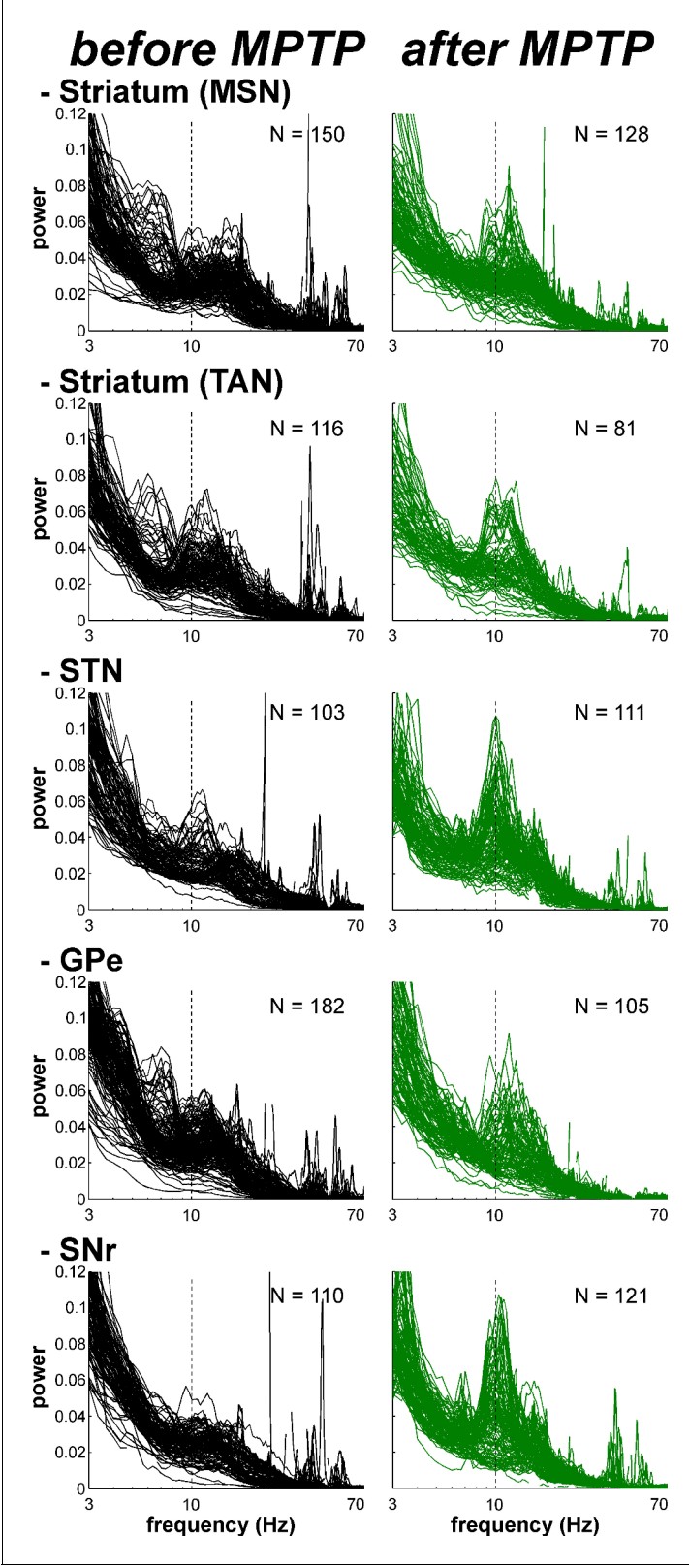

**Figure 12.** Exaggerated 8–15 Hz oscillations of LFPs within all structures of the BG network in parkinsonism. Superimposed PSDs of the LFPs recorded in the different BG nuclei (Striatum, STN, GPe and SNr) in the normal (before MPTP) and parkinsonian (after MPTP) state. A custom-made artifact removal procedure was used to clean sharp peak artifacts recorded by the high impedance microelectrodes. Abscissas are in log scale. N is the number of recording sites.

*Figure 12 continued on next page*

*Figure 12 continued*

The following source data is available for figure 12:

**Source code 1.** Custom-made artifact removal procedure.
**Source data 1.** Power spectral densities of the local field potential before and after MPTP.

(*Figure 14*, right polar histograms) and multi-unit (*Figure 14—figure supplement 1*, right polar histograms) spiking activity of TAN, STN, GPe and SNr neurons mainly ranged from 180 to 270°. Namely, TAN, STN, GPe and SNr neurons tended to fire around the negative peak of the 8–15 Hz oscillatory LFP - but see *Nelson and Pouget (2010)* for possible confounding effects of electrode and instrument filter properties on the exact phase relationships between spikes and LFPs. We verified the results obtained with the coherence/phase analysis by examining the population spike-triggered averages of the LFP (STAs LFP). The STA LFP analysis revealed that except for MSNs, the coincidence between the spikes of the STN, GPe, SNr neurons, as well as those of the TANs, and the negative waves of the LFP oscillations increased after MPTP intoxication (*Figure 16*, left and central columns). Finally, to rule out the possibility that differences in the population STAs LFP were not a mere consequence of the differences in discharge rate (and the total number of triggers) of the different BG neuronal components (*Figure 17*), we also calculated the STAs LFP after random dilution of the spike trains of the neurons (see Materials and methods). We found that the random dilution of the spike trains did not affect the STA-LFP results (*Figure 16*, right column). Therefore, in the parkinsonian state, spike-LFP synchronization was not influenced by the firing rate of BG neuronal components and more importantly, the absence of phase locking between MSN spiking activity and striatal LFPs was probably not due to the confounding effects of the slow discharge rate of the MSNs.

## Discussion

### Persistent population modulations of activity along the BG main axis support state-to-action coupling

The BG main axis connects the brain areas encoding the current state of the subject and the brain areas controlling action. However, current BG computational models do not explicitly describe how the BG main axis (actor) holds and evaluates transient sensory information (i.e., current state) until it can be used later by the cortical and brainstem motor centers that guide the optimal action/response. Therefore, it is crucial to show that persistent modulatory activity (i.e., during prolonged delays between sensory perception and action, especially when sensory information is no longer available) is not a unique property of the frontal cortex and other cortical areas (*Curtis and Lee, 2010*; *Durstewitz et al., 2000*; *Goldman-Rakic, 1995*), but can also be found along the BG main axis.

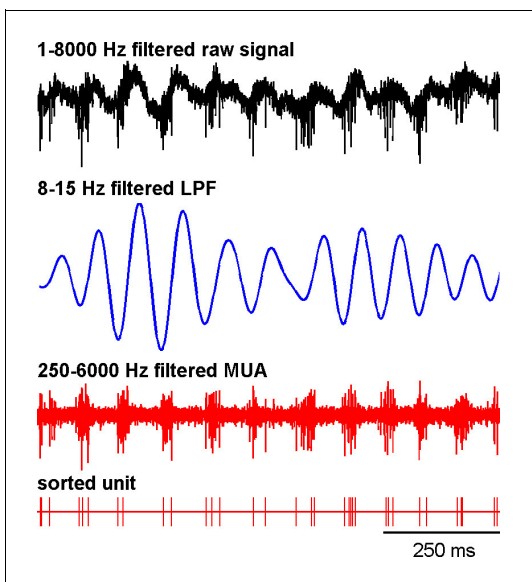

**Figure 13.** Recording of the subthalamic spiking and LFP activity after MPTP intoxication. The traces reflect the different filter properties applied to the signal recorded by the electrode. Upper trace depicts the broad (1–8000 Hz) band-pass filtered trace. The second trace is band-pass filtered at the low beta range (8–15 Hz) and depicts LFP oscillations. In the third trace, a 250–6000 Hz band-pass filter is used and reveals the multi-unit activity (MUA) that exhibits a periodic oscillatory pattern synchronized to the 8–15 Hz LFP oscillation. Below the 250–6000 Hz filtered MUA is the digital display of the spikes detected online from the MUA using the online template-matching algorithm.

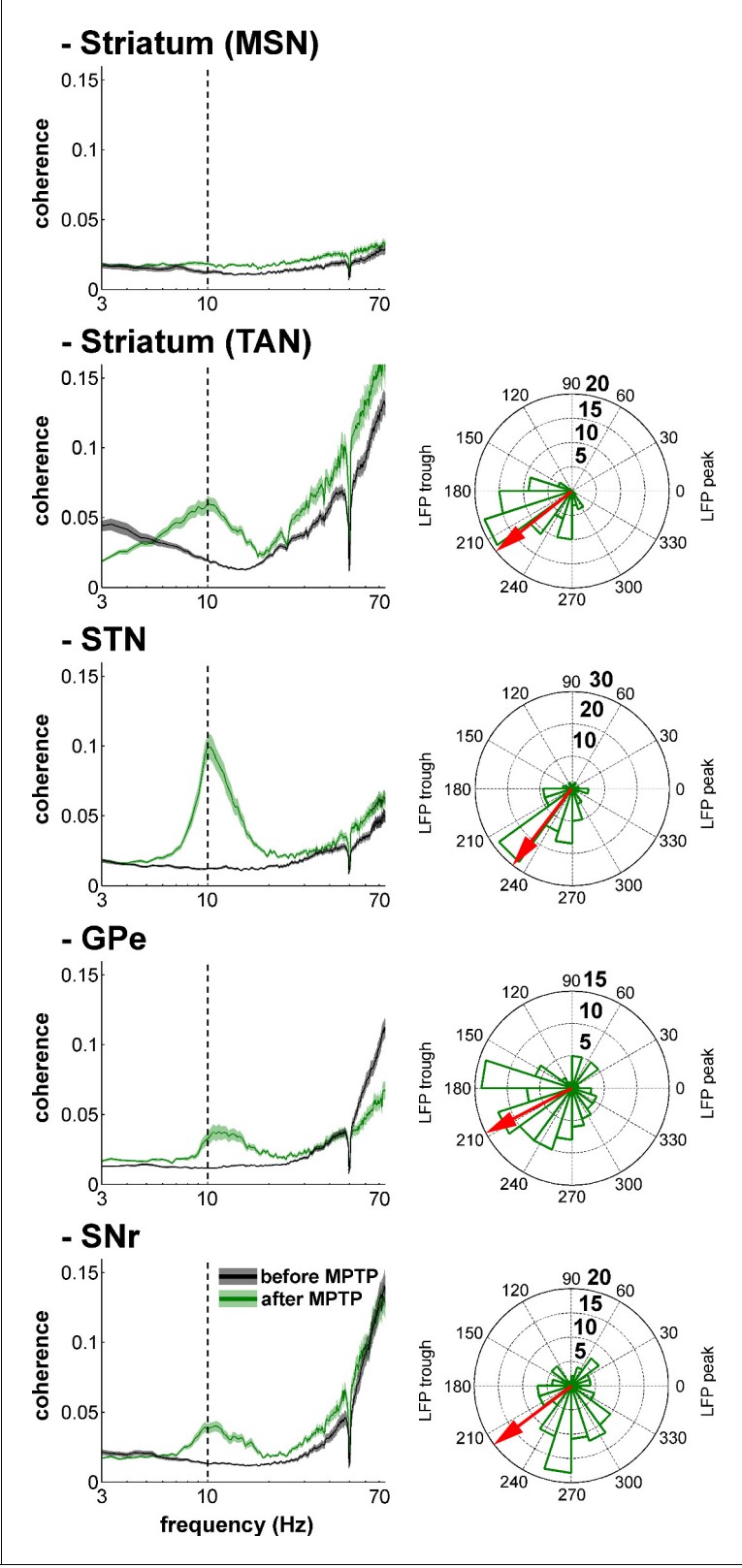

**Figure 14.** Single-unit spiking activity and LFP synchronization for the BG neuronal assemblies before and after MPTP intoxication. Left, average spike-field coherences between the spike train of the neurons (single-unit spiking activity) and the LFPs recorded in their vicinity before and after MPTP. Abscissas are in log scale. Right, phase histograms of the phases of the single-unit spiking activity relative to the LFP at the frequency of coherence peak values calculated in the 8–15 Hz range, after MPTP intoxication. The red arrows represent the mean phase of the spiking activity relative to the

*Figure 14 continued on next page*

*Figure 14 continued*

LFP peak (phase = 0 degree). Numbers in bold indicate the numbers of spike-LFP pairs. Given the absence of spike-field coherence for the MSNs after MPTP intoxication, the phase histogram of the MSN spike-LFP pairs was not calculated.

The following source data and figure supplement are available for figure 14:

**Source data 1.** Spike-LFP synchronization before and after MPTP.

**Figure supplement 1.** Multi-unit spiking activity and LFP synchronization for the BG neuronal assemblies before and after MPTP intoxication.

Unlike the cortex, many projection neurons of the BG use the inhibitory transmitter GABA as a carrier of information within and between BG nuclei. Nevertheless, the STN is a major source of glutamate (an excitatory transmitter) to the BG downstream structures (GPe, GPi and SNr). Accordingly, in most BG structures, modifications of spiking activity by behavioral events consist of either increases or decreases in firing rate (*Espinosa-Parrilla et al., 2013*; *Joshua et al., 2009b*; *Turner and Anderson, 2005*). Due to their low baseline firing rate (*Figure 17A*), it might be posited that like other low-discharge rate neurons (e.g., in the cortex), the neuronal responses of the striatal MSNs would be restricted to increases in activity. Nevertheless and in line with earlier studies (*Adler et al., 2012*; *Báez-Mendoza et al., 2016*; *Samejima et al., 2005*), we found task-related decreases (albeit less frequent than increases) in the discharge rate of MSNs. This suggests that neurons in the BG main axis (including striatal MSNs) exhibited diverse timing and directional (increase or decrease) patterns of discharge modulations from cue onset to outcome delivery.

These diverse patterns of BG discharge modulations resulted in persistent population activity, even in the absence of sensory stimulation (after cue offset) (*Figures 3*, *4* and *5*). Hence, in each neuronal assembly of the BG main axis, a modulation of the population activity may form a persistent internal representation (*Curtis and Lee, 2010*; *Durstewitz et al., 2000*; *Goldman-Rakic, 1995*). Our working hypothesis holds that the BG persistent population activity reflects the neuronal coupling between the current state (defined by cue presentation) and the upcoming contingent behavioral response (licking or blinking movement in our task). The heterogeneity of the BG main axis responses (*Adler et al., 2012*; *Deffains et al., 2010*; *Joshua et al., 2009a*) contrasts with the homogeneity of the transient responses of the BG neuromodulators (*Graybiel et al., 1994*; *Joshua et al., 2009a*) and corroborates the view that the BG main axis has a large information capacity that can encode the enormous and diverse possibilities of state-to-action mapping (*Bar-Gad et al., 2003*).

## Subthalamic rather than striatal activity fluctuations correlate with the time-varying increase-decrease balance of spiking activity in BG downstream structures

For many years, the STN was considered to be a relay station of the BG indirect pathway (striatum-GPe-STN-GPi/SNr) under the exclusive influence of cortico-striatal transmission, whereas the role of the (hyper) direct projections from the cortex to the STN (*Haynes and Haber, 2013*; *Nambu, 2004*) was neglected (*Bergman et al., 1990*). Here, we found that in the normal awake monkey engaged in a temporal discounting classical conditioning task, the fluctuations in STN activity were concurrent with the modifications of activity in the BG downstream network, without apparent parallel modulations of striatal activity (*Figures 5* and *6*). Our experimental methods (i.e., extracellular recording of spiking activity in behaving animals) did not allow us to differentiate between the activity of MSNs of the BG direct (D1 MSNs-GPi/SNr) and indirect (D2 MSNs-GPe-STN-GPi/SNr) pathways (*Albin et al., 1989*; *Gerfen et al., 1990*; *Figure 2A*). However, examination of the temporal patterns of changes in activity in the MSN assembly did not indicate the presence of two distinct sub-populations (*Figures 3*, *4*, *5* and *6*), which is consistent with recent studies revealing the co-activation of the striatal direct and indirect pathways (*Cui et al., 2013*; *Oldenburg and Sabatini, 2015*). In addition, a formal clustering analysis of the MSN responses from cue onset to outcome delivery in both conditions (data not shown) found no evidence for two distinct MSN sub-groups (e.g., one group that responded during cue presentation and another group that responded during the extended delay period) that could account for the reversal of I/D balance in the BG downstream network during the delay period in the delayed condition.

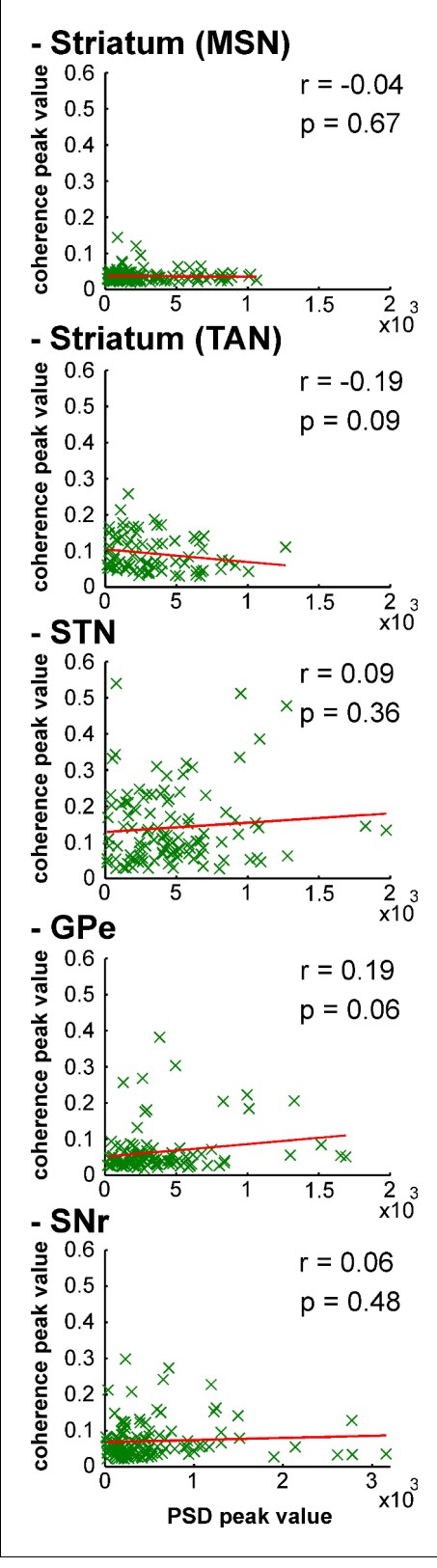

**Figure 15.** Strength of LFP oscillations is not related to the degree of synchronization between single-unit spiking activity and LFP oscillations along the BG network after MPTP intoxication. Scatter plots of the 8–

Although there is an ongoing debate on the distribution of STN output in the BG downstream network (*Mathai and Smith, 2011*; *Sato et al., 2000*), a compelling functional model of action selection process in the BG network claims that information transfer via the hyper-direct (cortex-STN-GPi/SNr) pathway is faster and more broadly distributed in the BG output structures than the information flowing along the direct and indirect (striatum-GPi/SNr) pathways (*Mink, 1996*; *Nambu et al., 2002*). Given that the striatum and the STN are the two input structures of the BG network and that STN neuronal responses were more strongly correlated with BG downstream neuronal responses, compared to striatal MSN responses (*Figure 7B*), we suggest that the STN might orchestrate the time-varying I/D balance in the BG downstream network. Thus, the STN would exert a powerful glutamatergic and apparently GABAergic striatal-free control over the release of BG output commands (*Baunez et al., 2001*; *Frank et al., 2007a*; *Jahanshahi et al., 2015*) prior to their fine modulation by the direct and indirect pathways. Nonetheless, this specific correlation between the divergent excitatory STN projections and BG downstream neurons was only shown here for a classical conditioning task with immediate and delayed outcomes and should be tested in the future under a range of behavioral conditions and paradigms.

## Abnormal oscillatory activity is not shared by all BG neuronal components in parkinsonism

In this study, we used a harsh MPTP intoxication procedure (see Materials and methods) rather than a slower regimen of MPTP intoxication (i.e., that would have led to a L-DOPA responsive parkinsonian monkey capable of executing behavioral tasks). This decision was made to induce a large striatal dopamine depletion (*Figure 8*) and guarantee the emergence of the full spectrum of pathological synchronous oscillations in the BG. Indeed, earlier studies have failed to find abnormal BG oscillatory activity at the early stage of chronic MPTP treatment (*Leblois et al., 2007*). In addition, the stability of the parkinsonian symptoms during the complete duration of the recordings was critical in our study (*Figure 9A*) to enable the comparison of activities at the different stages of the BG. Therefore, we favored severe MPTP-treated monkeys that usually develop stable clinical symptoms (*Potts et al., 2014*).

*Figure 15 continued*

15 Hz LFP PSD peak values and the 8–15 Hz spike-field coherence peak values for the different BG neuronal components. Red line represents the linear regression line between the LFP PSD peak values and the spike-field coherence peak values. r is the Pearson correlation coefficient and p indicates the probability that r = 0.

The following figure supplement is available for figure 15:

**Figure supplement 1.** Strength of LFP oscillations is not related to the degree of synchronization between multi-unit spiking activity and LFP oscillations along the BG network after MPTP intoxication.

The emergence of synchronized oscillatory activity in the BG network of animal models of PD and human PD patients has been well-described (*Brown, 2003*; *Wichmann and DeLong, 2003*). These abnormal oscillations are thought to alter the information flow through the BG main axis, thus leading to the release of abnormal commands by BG output structures (*Bergman et al., 1998*; *Brown et al., 2001*). However, the frequency range of these oscillations varies across species (*Stein and Bar-Gad, 2013*). In line with earlier studies conducted on MPTP-treated monkeys (*Bergman et al., 1994*; *Raz et al., 2001*; *Soares et al., 2004*), we found that abnormal spiking oscillatory activity after MPTP intoxication of African green monkeys (vervets) occurred in the low beta (8–15 Hz) range. In addition, we showed that this pathological oscillatory activity, at least at the level of spiking activity, was not shared by all BG neuronal components in parkinsonism (*Figures 10* and *11*). Even though earlier rodent (*Costa et al., 2006*) and primate (*Singh et al., 2015*) studies have shown that MSN activity patterns are altered in parkinsonism, we found that MSNs apparently failed to express pathological 8–15 Hz neuronal oscillations, as did the STN and BG downstream structures (*Figures 10* and *11*). The lack of oscillatory activity of MSNs in the MPTP-treated monkey is consistent with our unpublished observations of no oscillatory spiking activity in the striatum of parkinsonian human patients undergoing DBS procedures. However, not all components of striatal activity were deprived of abnormal oscillatory activity after MPTP intoxication. Indeed, as already described in earlier studies of TANs in MPTP-treated monkeys (*Raz et al., 1996*), we found that TANs also exhibited 10 Hz oscillatory activity. This MSN-TAN incongruity is in line with the view of a weak functional in vivo connectivity between MSNs and TANs (*Adler et al., 2013b*) and probably reflects their different synaptic drives.

## Abnormal oscillations in parkinsonism resonate across the BG network through the STN, not the striatum

In each structure of the BG main axis, including the striatum, we found exaggerated 8–15 Hz oscillations of LFP after MPTP intoxication (*Figure 12*). Nevertheless, 8–15 Hz LFP oscillations (although sparser) were also observed in the healthy state (before MPTP). In normal behavior control, beta oscillatory activity is known to play a role in the maintenance of the current sensorimotor or cognitive state (the status quo) (*Brittain and Brown, 2014*; *Engel and Fries, 2010*). Therefore, the 8–15 Hz LFP oscillations observed in the healthy state likely correspond to one of the neuronal processes involved in state-to-action mapping. Consequently, exaggeration of 8–15 Hz oscillatory activity after induction of parkinsonism would compromise the ability of human PD patients to adjust their behavior to situational demands (*Brown and Marsden, 1988*; *Cools et al., 1984*; *Frank et al., 2007b*) due to an abnormal persistence of the BG 'status quo' signal.

The LFP probably represents a global signature of the synaptic input and dendritic processing, whereas MUA reflects the efferent (output) activity of the local neuronal population (*Buzsáki et al., 2012*; *Logothetis, 2003*). Thus, as for other structures in the closed loop of the BG network, we found that the dopamine-depleted striatum was also bombarded with pathological oscillations by its major afferent neurons (*Figure 12*). However, these striatal oscillating synaptic inputs apparently only entrained the spiking activity of the TANs, but not the MSNs (*Figures 10* and *11*). Spike-field coherence analysis revealed that the single-unit (*Figure 14*) and multi-unit (*Figure 14—figure supplement 1*) spiking activities of the STN, GPe and SNr neurons after MPTP intoxication were synchronized to the 8–15 Hz oscillatory LFP recorded in their surrounding areas. On the other hand, in the dopamine-depleted striatum, only TAN (single- and multi-unit) spiking activity was phase locked to the striatal 8–15 Hz LFP oscillations. After MPTP intoxication, we found no evidence for a (linear) relationship between the strength of the 8–15 Hz LFP oscillations and the degree of spike-field coherence in the 8–15 Hz range (*Figure 15* and *Figure 15—figure supplement 1*) along the BG

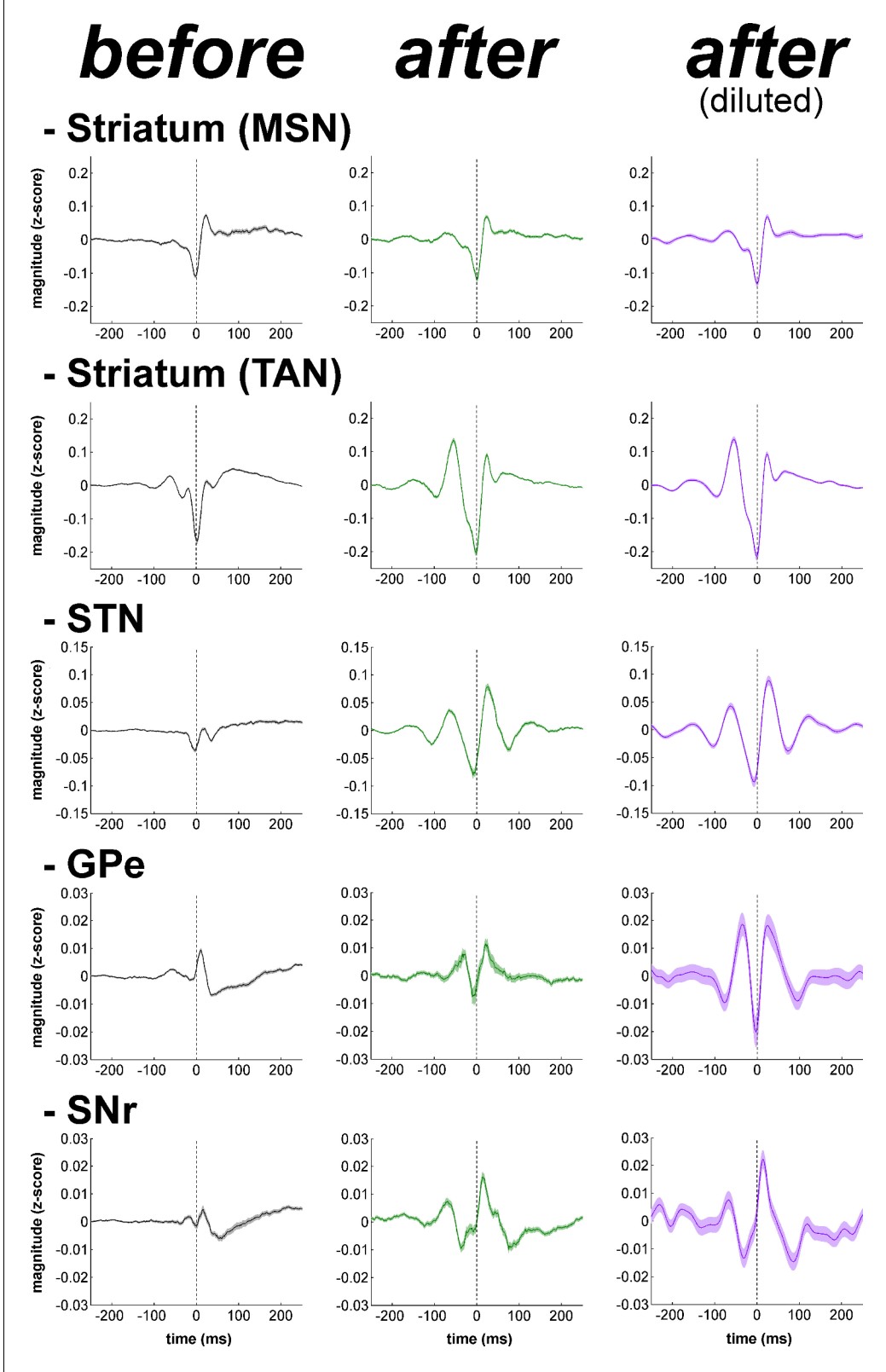

**Figure 16.** Population spike-triggered averages of LFP after MPTP intoxication show larger fluctuations around spike times of STN, BG downstream neurons and striatal TANs, but not around spike times of striatal MSNs. For each BG neuronal assembly, population STA LFP was defined as the average of the STAs LFP of all neurons. After MPTP intoxication, STAs LFP were calculated before and after random dilution of the spike trains of the

*Figure 16 continued on next page*

*Figure 16 continued*

neurons. LFP was recorded in the vicinity of the neuron (i.e., spiking activity and LFP were recorded on the same electrode). Dashed vertical lines indicate the time of the spikes (time = 0). The shaded areas mark SEMs

network. Therefore, whatever the strength of the 8–15 Hz LFP oscillations within the striatum, MSN spiking activity failed to lock with these LFP oscillations.

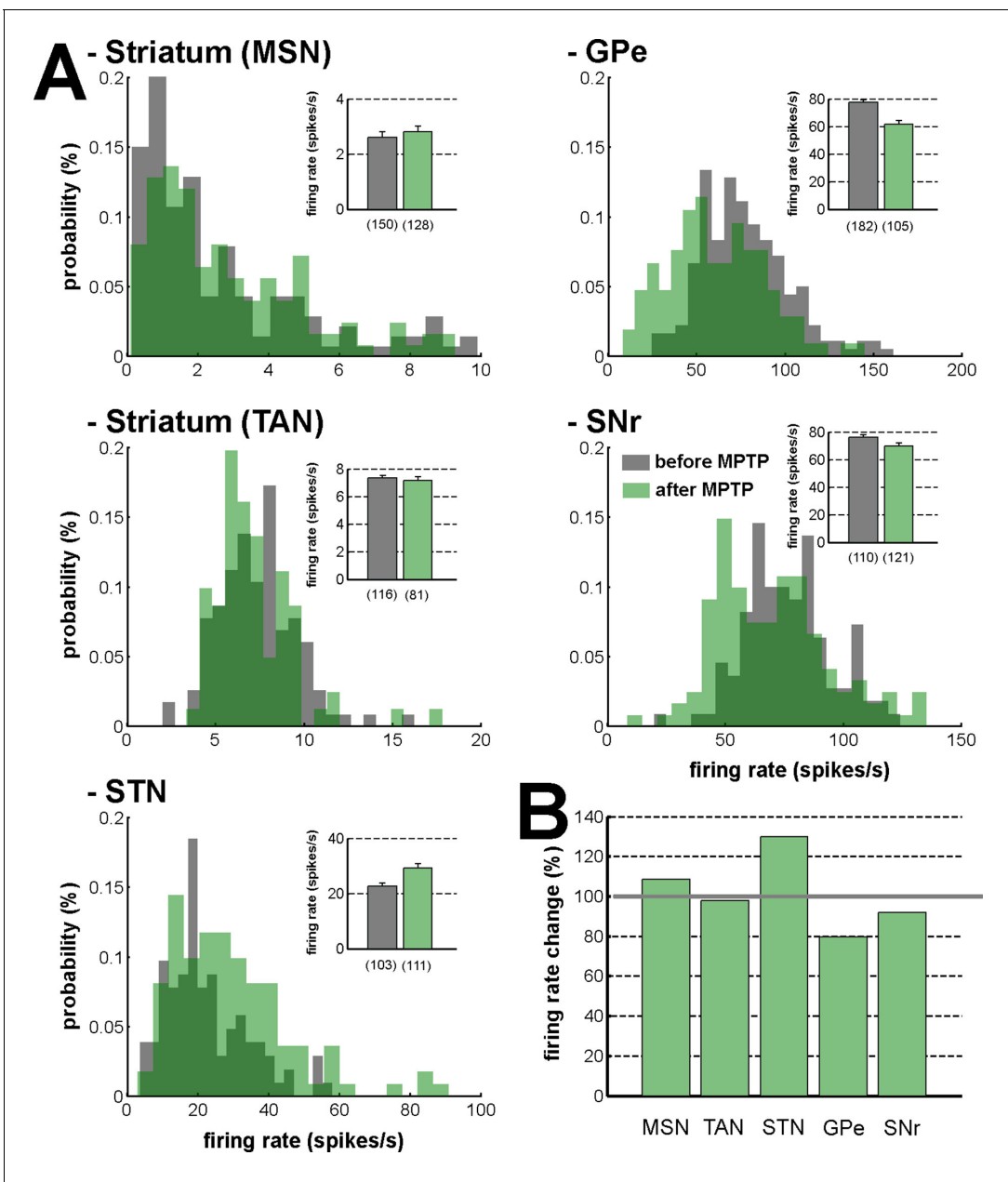

**Figure 17.** Firing rate in the BG network before and after MPTP intoxication. (**A**) Distribution of the firing rate of the neuronal components of the BG network. Gray - before MPTP; light green - after MPTP; dark green - overlapping bins. Inset, mean firing rate before and after MPTP intoxication. Error bars represent SEMs. Numbers in parentheses indicate the number of neurons. (**B**) Percentage of change in the firing rate after MPTP intoxication, for the neuronal components of the BG network. Gray line (100%) represents the healthy (before MPTP) firing rate.

Unlike the low discharge rate of the isolated MSNs (lower than the frequency of the 8–15 Hz oscillations, *Figure 17*), the MUA recorded in the vicinity of the isolated striatal MSNs reflects the discharge of many neurons. Therefore, given the consistency of the spike-field coherence results using spike trains of isolated MSNs (*Figure 14*) or striatal MUAs (*Figure 14—figure supplement 1*), the absence of coherence between MSN spiking activity and LFP after MPTP intoxication was most likely not confounded by the slow MSN discharge. Indeed, STA LFP analysis (even after random dilution of the spike trains of the neurons) also indicated that MSN spikes did not synchronize with LFP oscillations, as did the spikes of TAN, STN, GPe and SNr neurons (*Figure 16*). Finally, in line with earlier studies (*Goldberg et al., 2004*; *Kühn et al., 2005*), we showed that these synchronized spiking activities occurred during the negative wave of the 8–15 Hz oscillatory LFP (*Figures 14*, *16* and *Figure 14—figure supplement 1*). Together, these results rule out the possibility that the absence of synchronization between MSN spiking activity and abnormal oscillating LFP striatal inputs in parkinsonism was due to the low discharge rate of the MSNs. Rather, the abnormal striatal synaptic inputs may not have been strong enough to depolarize the membrane potential of the MSNs above the spike threshold.

Based on the classical rate model of the BG network (*Albin et al., 1989*; *Bergman et al., 1990*; *Gerfen et al., 1990*), striatal dopamine depletion leads to a reduction of the D1-MSM discharge and an elevation of the D2-MSN discharge. In contrast to an earlier study (*Liang et al., 2008*), we did not find any robust increase in the firing rate of the MSNs recorded within the dopamine-depleted striatum (*Figure 17*). This inconsistency could be due to differences in the experimental approaches, such as the species, the methods of MPTP intoxication, the time elapsed between MPTP treatment and neuronal recordings, and the severity of the parkinsonian symptoms. In any case, our methods of extracellular recording in behaving animals cannot discriminate between D1- and D2-MSNs. However, in agreement with earlier studies (*Bergman et al., 1994*; *Filion and Tremblay, 1991*; *Miller and DeLong, 1987*), we also found that after MPTP intoxication, the firing rate decreased in the GPe and increased in the STN (*Figure 17*). Therefore, we cannot rule out the possibility that the slight elevation of the striatal GABAergic drive to the GPe (via D2-MSNs, *Figure 2A*) may have induced oscillatory activity in the BG downstream structures (*Terman et al., 2002*), probably through changes in the activity pattern of the GPe-STN closed circuit (*Bevan et al., 2002*; *Plenz and Kital, 1999*).

Noticeably, despite the abnormal oscillating synaptic inputs in the striatum, striatal projection neurons (MSNs) failed to express spiking (output) activity. Therefore, abnormal oscillatory activity within dopamine-depleted striatum did not propagate to the BG downstream structures. In this study, we did not assess the striatal vs. extra-striatal dopamine depletion after MPTP intoxication (*Figure 8*) and we cannot exclude the possible role of dopamine degeneration of the STN and BG downstream structures in the emergence of oscillatory activity in the STN and along the BG network (*Francois et al., 2000*; *Galvan et al., 2014*; *Rommelfanger and Wichmann, 2010*). Nevertheless, the abnormal oscillatory spiking activity of the STN neurons emerged in tandem with an increase in synchronization between the oscillatory spiking activity of the STN neuronal targets (i.e., BG downstream neurons) and their abnormal LFP oscillations (i.e., BG downstream oscillating synaptic inputs). Therefore, our results are in line with other studies (*Nambu and Tachibana, 2014*) and suggest that the abnormal oscillations observed in PD resonate across the closed loop of the cortico-BG network through the STN, not the striatum.

## Concluding remarks

Overall, the current study conducted on both normal and parkinsonian animals support the idea that the STN may be the driving force behind the physiology (*Kitai and Kita, 1987*) as well as the pathophysiology of the BG. Nevertheless, the striatum obviously remains a major player in BG functioning in health and disease. Striatal dopaminergic input is involved not only in the modulation of the efficacy of the cortico-striatal synapses (*Reynolds et al., 2001*; *Shen et al., 2008*), but also in the excitability of the striatal projection neuron (*Nicola et al., 2000*; *Onn et al., 2000*). Thus, the coincidence of cortical/thalamic and midbrain dopaminergic activity might be a necessary condition for MSN spike generation. Consequently, MSN spiking activity would be shaped by striatal inputs only during brief, but functionally important periods (probably marked by transient dopaminergic activity). This refined control of striatal output patterns was probably undetected in the current population analysis.

Moreover, in parkinsonism, due to the utilization of a harsh MPTP intoxication procedure, our MPTP-treated monkeys developed stable and severe motor symptoms, and were unable to execute the behavioral task. Therefore, the study of the neuronal activity along the BG network in similar classical (but also operant) conditioning tasks using MPTP-treated monkeys that exhibit moderate parkinsonian symptoms is undoubtedly a necessary step that should enhance our understanding of the physiology and pathophysiology of the BG. However, the unique anatomical and physiological features of the STN highlighted in the current study provide a concrete explanation for the efficacy of the inactivation of the STN in MPTP-treated monkeys (*Bergman et al., 1990*), human patients (*Alvarez et al., 2005*) and STN-DBS in MPTP-treated monkeys (*Benazzouz et al., 1993*) and parkinsonian patients to alleviate BG-related motor (*Odekerken et al., 2016*) and non-motor (*Lhommée et al., 2012*) clinical symptoms.

## Materials and methods

All experimental protocols were conducted in accordance with the National Institutes of Health *Guide for the Care and Use of Laboratory Animals* and with the Hebrew University guidelines for the use and care of laboratory animals in research, supervised by the Institutional Animal Care and Use Committee of the faculty of medicine, the Hebrew University, Jerusalem, Israel. The Hebrew University is an Association for Assessment and Accreditation of Laboratory Animal Care (AAALAC) internationally accredited institute.

### Animals and behavioral task

Two female African vervet green monkeys (K and S, *Cercopithecus aethiops aethiops)*, weighing ~4 kg, were trained in a temporal discounting classical conditioning task (*Figure 1A*). Six different fractal visual cues were presented for 2 s. The cues used here were full-screen isoluminant images generated using ChaosPro 3.2 program (www.chaospro.de) and displayed on a 21 inch LCD monitor, located 50 cm in front of the monkeys' faces. Cues predicted either a food outcome, an airpuff outcome (directed at both eyes) or neither of them, thus grouping them into three categories: appetitive/rewarding, aversive and neutral cues. For each category, depending on the cue, its offset was either immediately followed by the outcome period (immediate condition) or by a 6-s delay period which preceded the outcome period (delayed condition). This delay period was chosen on the basis of previous studies indicating that monkeys engaged in a delayed-release task maintain central fixation for no more than 4 s (*Slovin et al., 1999*) and that trial-over-trial behavioral learning of monkeys during smooth pursuit eye movement task is forgotten within 6 s (*Yang and Lisberger, 2010*, *2014*). The outcome period (0.15 s) was signaled by one of three sounds that differentiated the three possible outcomes (food, airpuff or no outcome) and was followed by a variable inter-trial interval (ITI) of 6–10 s. Fractal images and sounds were swapped between the two monkeys.

### Surgery and magnetic resonance imaging

After an intensive training period in the task (~3 months, 5–6 days a week, 300–600 trials/day), the monkeys underwent a surgical procedure for the implantation of a MRI-compatible Cilux head holder (Crist Instruments, MD) and a 27 mm (inner edge) square Cilux recording chamber (AlphaOmega Engineering, Israel). During surgery, the head of the animal was fixed in a stereotaxic frame (David Kopf Instruments, CA) and the recording chamber was implanted over the right hemisphere, above a burr hole in the skull. The head holder and the chamber were attached to the skull using titanium screws (Crist Instruments, MD) and wires (Fort Wayne metals, IN) embedded in acrylic cement. The recording chamber was tilted 45° laterally in the coronal plane and stereotaxically positioned to cover most of the basal ganglia (BG) nuclei (*Contreras et al., 1981*; *Martin and Bowden, 2000*).

Surgery was performed under aseptic conditions and isoflurane and N$_2$O deep anesthesia, following induction with Medetomidine (0.1 mg/kg IM) and Ketamine (10 mg/kg IM). The surgical procedure was carried out by a board-certified neurosurgeon and anesthesia induction and level maintenance (end-tidal CO$_2$, arterial O$_2$ saturation, ECG, blood pressure and rectal temperature) were performed by the veterinary team of the animal facility. Analgesics (Carprofen, 4 mg/kg SC) and antibiotics (Ceftriaxone, 35 mg/kg IV or IM) were administered during surgery and the first 2–3 postoperative days.

Recording began after a postoperative recovery period of 5–7 days. During this recovery period, an anatomical MRI scan (Whole-body 3T, 32-channel head coil, Siemens scanner) was performed to ascertain the correct placement of the chamber and estimate the stereotaxic coordinates of the neuronal recordings. The MRI sequences included a high-resolution T2-weighted TSE sequence: TR = 4232 ms, TE = 63 ms, flip angle = 180°. 70*1 mm coronal slices (no gap) were used with a field of view (FOV) of 240*240 mm and matrix 320*313 (in-plane resolution of 0.75*0.77 mm). Four averages were used to improve the signal to noise ratio. The coronal slices were positioned on a T2-weighted TSE sagittal sequence (TR = 2200 ms, TE = 63 ms, flip angle = 180°, 35 slices of 1 mm with 10% gap, FOV = 240*240 mm, matrix = 320*313). Before the MRI scan, the chamber was filled with 3% agar and five tungsten electrodes were inserted into the animal's brain at equally spaced coordinates of the recording chamber [Y,X = (6,0), (0,−6), (0,0), (0,6) and (−6,0) in mm from the chamber center]. An additional MRI scan was performed at the final stage of the recordings to re-validate the location of the recording sites and to rule out significant brain shifts. For the purpose of MRI examination, the animal was sedated using IM Medetomidine 0.1 mg/kg and IM Ketamine 10 mg/kg.

Throughout the entire course of the experiments, the chambers were washed 1–2 times/day with neomycin-sulfate saline solution followed by saline solution.

## Induction of experimental parkinsonism

To induce parkinsonism, the monkeys were treated with 1-methyl-4-phenyl-1,2,3,6-tetrahydropyridine (MPTP-hydrochloride, Sigma, Israel). Five IM injections of 0.35 mg/kg/injection were made over the course of four days (two injections in the first day) under Ketamine (10 mg/kg IM) sedation. We used a modified Benazzouz primate parkinsonism scale (*Benazzouz et al., 1995*) to assess the severity of the level of parkinsonism. Neuronal recordings during the parkinsonian state began when the animal exhibited severe parkinsonian symptoms, 7 days after the first injection (*Figure 9*).

Since following the MPTP injections both animals became severely akinetic and lost the ability to feed themselves, they were fed with a high-calorie nutritional supplement (Ensure Plus, Abbott Labs Nutrition, OH) through a nasogastric tube 2–3 times/day, 7 days/week. In addition, special care was taken such as the use of soft mattresses and frequent changes of position to prevent the development of pressure sores.

Two or three weeks after the first MPTP injection, we started dopamine-replacement therapy (Dopicar, Teva Pharmaceutical Industries, Israel; 125 mg L-DOPA + 12.5 mg Carbidopa twice a day). From then on, neuronal recordings were conducted while the animals were off dopaminergic medication (overnight washout >12 hr) to mirror the conditions of recording observed in human parkinsonian patients undergoing DBS surgery. In the subsequent hours of L-DOPA administration (at the end of the daily recording session), significant clinical improvement occurred in both animals, thus corroborating the diagnosis of parkinsonism. Nevertheless, during the neuronal recordings before (MPTP naive) and after (MPTP off medication) dopamine-replacement therapy, both animals exhibited severe bradykinesia, a reduction in spontaneous blinking frequency, marked rigidity, significantly flexed posture and episodic low (~4–6 Hz) frequency tremor (*Figure 9*). The data collected during the two states (MPTP naive and MPTP off medication) were grouped since no significant difference was detected between them (data not shown).

## Assessment of dopamine depletion in the MPTP-treated monkeys

F-18 FDOPA positron emission tomography (PET) scan and post-mortem tyrosine hydroxylase (TH) immunohistochemistry (Monkey K and S, respectively) were used to assess the severity of dopamine depletion following the MPTP treatment.

### Positron emission tomography scan

After completion of the neuronal recordings in the MPTP-treated monkey K, a dynamic PET of its brain was performed on a Discovery ST PET/CT system (GE Medical System, Milwaukee, USA). As a control, and in the same session using the same radioligand, another healthy female African vervet green monkey was scanned. The monkeys were anesthetized and positioned in the prone position in the PET/CT scanner. The dynamic PET scan started at the time of injection of 5.1 and 3.0 mCi (or 188.7 and 111 MBq, amount adjusted to the weight of the monkey) F-18 FDOPA (synthesized at the Cyclotron of the Hadassah Medical Center) for the MPTP-treated (Monkey K) and healthy monkeys,

respectively. For both monkeys, dynamic F-18 FDOPA PET was acquired with time frames as follows: 6x30 s, 7x1 min, 6x5 min, 5x10 min; total scan time was 90 min. PET scans were reconstructed with OSEM (ordered subset expectation maximization) iterative reconstruction.

Radioligand F-18 FDOPA accumulates in the synaptic terminal of the dopaminergic neurons. Therefore, for each monkey and during each time frame, the radioactivity concentration at the level of the striatum was normalized (Z-score normalization) by the radioactivity concentration in the region of interest (ROI) for the cerebellum (i.e., region of non-specific F-18 FDOPA uptake).

## Histology

After the last recording session, MPTP-treated monkey S was deeply anesthetized with a lethal dose of pentobarbital and perfused through the heart with saline, followed by a 4% paraformaldehyde fixative solution. As a control, a healthy macaque (*Macaca mulatta*) monkey (used for another anatomical study) was perfused in a similar way. Brains were removed and cryoprotected in increasing gradients of sucrose (10, 20, and finally 30% ). Adjacent serial sections of 50 µm, from both healthy and MPTP-treated monkeys, were processed for TH immunohistochemistry. Sections were incubated with antisera to TH (mouse anti-TH, 1:20,000; Eugene Tech, Allendale, NJ) in 0.1 m phosphate buffer with 0.3% Triton X-100 and 10% normal goat serum (Incstar, Stillwater, MN) for 4 nights at 4°C and further processed using the avidin–biotin method (rabbit Elite Vectastain ABC kit; Vector Laboratories, Burlingame, CA).

## Data collection and physiological recordings

During the recording sessions, the monkeys' heads were immobilized with a head-holder and eight glass-coated tungsten microelectrodes (impedance range at 1 kHz: 0.3–0.8 MΩ) were advanced separately (Electrode Positioning System, AlphaOmega Engineering, Israel) toward and through the targeted BG structures. The electrical activity was amplified by 5000, band-pass filtered from 1 to 8000 Hz using a hardware four-pole Butterworth filter and sampled at 25 kHz by a 12-bit (± 5V input range) Analog/Digital (A/D) converter (Multi Channel Processor, AlphaOmega Engineering, Israel). Spiking activity was sorted online using a template matching algorithm (Alpha Spike Detector, AlphaOmega Engineering, Israel). Up to three different units could be isolated from the same electrode and the detection timestamp of the sorted units was sampled at 40 kHz.

The animals' task performance was assessed by monitoring licking and blinking behavior. Licking movements were recorded using an infra reflection detector (Dr. Bouis Devices, Germany) directed toward the animal's mouth. The infrared signal was amplified by 6, filtered between 1 and 100 Hz by a four-pole Butterworth filter and sampled at 1.56 kHz (Multi Channel Processor, AlphaOmega Engineering, Israel). Blinking movements were monitored using infrared digital video cameras which recorded the animal's face at 50 Hz. Eye states (open or closed) were automatically determined based on the number of dark pixels in the eye area, using custom software (*Mitelman et al., 2009*).

Neuronal data, licking movements, task and video synchronization digital signals were stored by the data acquisition system (Alpha-Map, AlphaOmega Engineering, Israel) for further analysis. Video data were stored on a separate PC. A home-made synchronization protocol ensured the synchronization of the behavioral, neuronal and video data.

## Identification of physiological targets

The different BG neuronal assemblies were identified according to their stereotaxic coordinates (based on MRI and primate atlas data (*Contreras et al., 1981*; *Martin and Bowden, 2000*) and the real-time assessment of their electrophysiological features. In the striatum, MSNs were differentiated from TANs and the fast spiking neurons (FSIs, presumably GABAergic interneurons co-expressing parvalbumin) based on their spike shape, discharge rate and pattern (*Adler et al., 2013b*; *Berke et al., 2004*; *Deffains et al., 2010*; *Sharott et al., 2009*).

The GPe border was characterized by sudden high frequency discharge activity below the striatum. The border between GPe and GPi was identified based on the depth of the electrode within the pallidum, the detection of the pallidal border cells (characterized by their typical regular firing pattern and broad action potential) and the differences in activity patterns of GPe and GPi cells (*Bezard et al., 2001*; *DeLong, 1971*). In the GPe, neurons exhibit either a high-frequency discharge interrupted by pauses (HFD-P neurons) or a low-frequency discharge with occasional brief high

frequency bursts (LFD-B neurons). In contrast, in the GPi, nearly all the neurons exhibit a continuous (without pauses) high-frequency discharge.

Beyond the internal capsule (discernible by the high density of fibers), the STN and SNr were differentiated based on the depth of the electrode, the level of background activity (higher background activity in the STN due to higher neuronal density), the electrophysiological features (narrower spike shape and higher firing rate in SNr) of the cells (*Bergman et al., 1994*; *DeLong et al., 1985*; *Schultz, 1986*) and the detection of the surrounding neurons (e.g., neurons of the zona incerta for the STN, and neurons of the substantia nigra pars compacta for the SNr).

Spike detection and sorting were done online (see 'data collection and physiological recordings' section above). Each sorted unit was offline subjected to visual inspection of the stability of its firing rate over its recording span and the longest stable period was manually selected for further analysis while the rest of the data was discarded. Then, the quantification of its isolation quality (only for the stable period) was graded by calculating its isolation score (*Joshua et al., 2007*). The isolation score ranged from 0 (i.e., multi-unit activity) to 1 (i.e., perfect isolation). Only cells exhibiting a stable firing rate for $\geq$15 min ($\geq$9 min for MSNs and STN neurons) and an isolation score $\geq$0.7 ($\geq$0.6 for MSNs and STN neurons) were included in the database. The adjustment of the inclusion criteria for MSNs and STN neurons was due to the extremely small soma of the MSNs and the highly dense cellular structure of the STN which make cell stability and isolation difficult. Analysis of the subset of MSNs and STN neurons that fulfilled the same inclusion criteria of TANs, pallidal and nigral neurons (stability $\geq$15 min, isolation score $\geq$0.7) yielded similar results to those reported here.

## Time-varying increase-decrease balance of spiking activity

Neuronal activities for the different task cues were characterized by their relative and absolute peri-stimulus time histograms (PSTHs, *Figures 2D* and *3*). The PSTHs (i.e., 6 PSTHs per neuron) were calculated in 1 ms bins and smoothed with a Gaussian window (SD = 20 ms). For each smoothed PSTH, we calculated the relative deviation from the baseline (i.e., relative PSTH) by subtracting the baseline firing rate (i.e., mean firing rate in the last 500 ms of the ITI) from the PSTH (*Figure 3*). We also calculated the absolute deviation from the baseline (i.e., absolute PSTH). Given that this operation does not provide a natural zero baseline, the average of the absolute deviation from the baseline during the last 500 ms of the ITI was subtracted from this statistic (*Figure 3*).

Next, the relative PSTH was segmented into consecutive and non-overlapped 20 ms bins. A bin was considered responsive when the bin activity $\geq$2 or $\leq -2$ SD of the baseline firing rate (p<0.05, empirical 68-95-99.7 rule). To examine the time course of the modulations of activity, we calculated the fraction of responsive bins at each time bin (20 ms). The fraction of responsive bins was calculated for all neurons in one of the BG structures, from cue onset to outcome delivery in both conditions (immediate and delayed) and for the different trial types (appetitive, neutral and aversive). BG main axis neuronal responses to behavioral events are composed of either increases or decreases in discharge rate (*Espinosa-Parrilla et al., 2013*; *Joshua et al., 2009b*; *Turner and Anderson, 2005*). For this reason, at each time bin, we determined for each neuron and for each trial condition and type (immediate/delayed conditions and appetitive/neutral/aversive types): (1) the fraction of increases and decreases in activity out of the total number of responsive bins, depending on whether the bin activity $\geq$ 2 or $\leq$ - 2 SD of the baseline firing rate, respectively (*Figure 6—figure supplement 1*); (2) the mean magnitude of increases and decreases in activity (*Figure 6—figure supplement 2*); (3) the relative dominance of the increases and decreases in activity *[(increase - decrease) / (increase + decrease)]* after having weighted the fractions of increases and decreases by their respective magnitude (i.e., increase-decrease balance of spiking activity or I/D balance, *Figure 6*). This way, we examined the temporal evolution of the I/D balance over the different task periods in each neuronal assembly of the BG main axis. Note that for each condition (immediate/delayed) we tested each neuron three times (for appetitive, neutral and aversive trials). Grouping all trial types of a single neuron revealed similar results (data not shown). Finally, segmentation of the relative PSTH using 50 ms bins yielded the same qualitative results as those reported (data not shown).

## Similarity/dissimilarity of neuronal responses between BG input and downstream neurons

For each neuronal activity (i.e., binned relative PSTH) of the BG main axis, we determined the principal polarity (increase or decrease) of the neuronal response during each task period (*Figure 7A*). The neuronal response was defined as an increase or a decrease when at least 75% of the modulated bins exhibited increases or decreases in activity, respectively.

Then, for each task period and each trial type, we calculated the zero lag coefficient of the cross correlation function (i.e., similarity coefficient) between the neuronal activities of each BG input-downstream neuron pair. Neuron pairs were clustered according to the principal polarity of their neuronal responses. This yielded four distinct scenarios: neuron pairs exhibiting a similar (increase-increase or decrease-decrease) or opposite (increase-decrease or decrease-increase) polarity of their neuronal responses (*Figure 7B*). The similarity coefficient ranged from −1 (i.e., opposite neuronal activity) to +1 (i.e. same neuronal activity). The similarity coefficients for the three trial types (appetitive, neutral and aversive trials) were pooled since no significant difference was detected between them.

## Power spectral density

For the power spectral density (PSD) calculations, the raw signal (1 to 8000 Hz, multi-unit activity, MUA) recorded in the vicinity of the cells that fulfilled the inclusion criteria of this study (stable discharge rate duration ≥15 or 9 min and isolation score ≥0.7 or 0.6 for TANs, pallidal, nigral neurons and MSNs, STN neurons, respectively) was online band-pass filtered from 250 to 6000 Hz (four-pole Butterworth filter). The filtered signal was Z-score normalized to obtain an unbiased estimate (by the electrode impedance or the amplitude of the recorded neuronal activity) of the oscillatory activity (*Zaidel et al., 2010*). The Z-normalized signal was rectified by the 'absolute' operator (*Deffains et al., 2014*; *Moran et al., 2008*; *Zaidel et al., 2010*). The rectified signal follows the envelope of the MUA and therefore enables the detection of burst frequencies below the range of the online band-pass filter (250–6000 Hz). Since the local field potential (LFP) frequency domain was filtered out, the resulting PSD only represented the oscillatory features of the spiking activity.

The PSD of each signal was calculated using Welch's method with a 3-s Hamming window (50% overlap) and a spectral resolution of 1/3 Hz (nfft = 75000, sampling frequency = 25,000 Hz). Any DC component generated by the sampling processes and by the 'absolute' operator was removed by subtracting the mean of every windowed segment of the rectified signal. For each neuronal assembly, the mean PSD was defined as the average of the PSDs of all signals (*Figure 10B*).

Quantification of the peak value of every PSD between 8–15 Hz was carried out by using a Z-score method (*Moshel et al., 2013*). Briefly, for every PSD, the peak value (maximum value) of the PSD between 8 and 15 Hz was measured, and a *tail* mean and standard deviation was defined in the frequency range of 55 to 75 Hz. In this *tail* range, no particular power spectrum phenomena were observed between before and after MPTP intoxication (*Figure 10*). The PSD peak value in the 8–15 Hz range was defined as the Z-score amplitude of the PSD peak value (*Figure 11*).

Similarly, we also calculated the PSD of the LFP (1 to 250 Hz) recorded in the vicinity of the isolated cells (*Figure 12*). To do so, the PSD of each Z-normalized LFP was calculated using Welch's method (see above for the parameters; but nfft = 2344, sampling frequency = 781.3 Hz) and without prior rectification by the absolute operator. Given the use of high-impedance microelectrodes, the LFPs were often contaminated by noise, which manifested as sharp peaks in the PSD. For this reason, we applied custom-made artifact removal software on the PSD of each LFP. For each peak of the PSD (PSD fpeak), if $PSD\ fpeak > 2*[(PSD\ fpeak_{-2} + PSD\ fpeak_{+2}) / 2]$, where fpeak was the frequency of the peak, PSD fpeak was considered as an artifact and PSD $fpeak_{-2}$, PSD $fpeak_{-1}$, PSD fpeak, PSD $fpeak_{+1}$ and PSD $fpeak_{+2}$ were removed (replaced with Matlab Not a Number, NaN).

## Spike-field coherence

For each sorted unit that fulfilled the inclusion criteria of the study (*Table 1*), the spike train (i.e. time epochs of the detected spikes) was low-passed filtered (cutoff frequency = 100 Hz) and Z-normalized. Synchronization between the Z-normalized spiking activity and the Z-normalized LFP recorded in the vicinity of the sorted unit was done by using the magnitude squared (MS) coherence method (*Figure 14*, left plots). Welch's method was utilized (see PSD calculation for the parameters).

Coherence values ranged (by definition) from 0 to 1. For every coherence estimate, we determined the coherence peak value (the highest coherence) in the 8–15 Hz range.

We also calculated the cross power spectrum density for all spike-LFP pairs. For each, we computed the cross spectrum phase and determined the phase lag value at the frequency of the 8–15 Hz coherence peak value. Then, for every neuronal assembly, we built the phase histogram of the phases of the spiking activity relative to the LFP at the frequency of the 8–15 Hz coherence peak values (*Figure 14*, right polar histograms).

To examine the relationship between the strength of the LFP oscillations and the degree of spike-LFP phase locking in the 8–15 Hz range, we performed a linear regression analysis between the 8–15 Hz LFP PSD peak values (see calculation of the 8–15 Hz MUA PSD peak values for the details) and the 8–15 Hz spike-field coherence peak value (*Figure 15*). Finally, we carried out a similar spike-field coherence analysis using the MUA rather than the spike train (single-unit activity) of the sorted units (*Figure 14—figure supplement 1* and *Figure 15—figure supplement 1*).

### Spike-triggered average of the LFP

To investigate the spike-LFP relationship in the temporal domain and not only in the frequency domain (i.e., spike-field coherence analysis), we also calculated the spike-triggered average (STA) of the LFP (*Goldberg et al., 2004*) for each sorted unit included in the neuronal database. To do so, each Z-normalized LFP was offline low-pass filtered (four-pole Butterworth filter, filtfilt Matlab function, cutoff frequency = 30 Hz). This low-pass filter removed the ringing artifacts following spike time and the residual waveform of the spikes in the LFP recorded in the vicinity of the sorted unit (i.e., from the same electrode). For each neuronal assembly, the population STA LFP was defined as the average of the STAs LFP of all neurons (*Figure 16*).

Finally, in the parkinsonian state, to rule out the possibility that STA LFP results in the striatum were influenced by the slow discharge rate of the MSNs (*Figure 17*), we randomly diluted the spike trains of every sorted units (MSNs included) firing at a higher rate than the mean discharge rate of the MSNs. Following the random dilution, the discharge rate of these units equaled the mean discharge rate of the MSNs (*Figure 17*) and then we repeated the STA-LFP analysis (*Figure 16*, right column).

### Statistics

Analysis was conducted identically on the different neuronal assemblies. The data from the two monkeys were pooled since no significant difference was detected between them. All statistical analyses were done using custom-made MATLAB 7.5 routines (Mathworks, Natick, MA, USA). Statistical comparisons between the different task periods were tested with one-way ANOVAs. If necessary, Bonferroni correction was used to adjust the chosen significance level according to the number of multiple comparisons. In addition, differences in fractions of neuronal responses depending on their principal polarity (increase vs. decrease) were assessed by using $\chi^2$ test and two-way ANOVAs were used to statistically compare the similarity coefficients between BG input (striatum or STN) and downstream (GPe, GPi or SNr) structures. Finally, for every neuronal assembly, two-sample t-test was used for statistical comparisons between the healthy (before MPTP) and parkinsonian (after MPTP) states. The criterion for statistical significance was set at $p<0.05$.

### Acknowledgements

We thank Dr. Yaron Dagan for assistance with animal care, Dr. Atira Bick and Dr. Nanette Freedman for assistance with the coordination and execution of the MRI and PET scans, and Esther Singer for editing. We also thank Anatoly Shapochnikov for help in preparing the experimental set up and Dr. Hila Gabbay and Dr. Sharon Freeman for general assistance. This work was supported by grants from the Edmond and Lily Safra Center (ELSC) to MD and LI; the Israel Ministry of Absorption and the Teva National Network of Excellence in Neuroscience (NNE) to LI; the Israel-US Binational Science Foundation (BSF) and the Adelis Foundation to ZI, SNH and HB; the European Research Council (ERC), the Israel Science Foundation (ISF), the German Israel Science Foundation (GIF), the Canadian Friends of the Hebrew University, the Rosetrees and Vorst Foundations and the Simone and Bernard Guttman Chair in Brain Research to HB.

# Additional information

## Funding

| Funder | Grant reference number | Author |
|---|---|---|
| The Edmond and Lily Safra Center | | Marc Deffains<br>Liliya Iskhakova |
| Ministry of Aliyah and Immigrant Absorption | | Liliya Iskhakova |
| The Teva National Network of Excellence in Neuroscience | | Liliya Iskhakova |
| The Israel-US Binational Science Foundation | | Suzanne N Haber<br>Zvi Israel<br>Hagai Bergman |
| The Adelis Foundation | | Suzanne N Haber<br>Zvi Israel<br>Hagai Bergman |
| European Research Council | GA 322495 CLUE-BGD 098777 | Hagai Bergman |
| Israel Science Foundation | | Hagai Bergman |
| The German Israel Science Foundation | I-1222-377.13/2010 002223 | Hagai Bergman |
| The Canadian Friends of the Hebrew University | | Hagai Bergman |
| The Rosetrees and Vorst Foundations | ROSETREES 251112 | Hagai Bergman |
| The Simone and Bernard Guttman Chair in Brain Research | | Hagai Bergman |
| The Rosetrees and Vorst Foundations | ROSETREES TRUST 271010 | Hagai Bergman |

The funders had no role in study design, data collection and interpretation, or the decision to submit the work for publication.

## Author contributions

MD, Conception and design, Acquisition of data, Analysis and interpretation of data, Drafting or revising the article; LI, SK, Acquisition of data, Drafting or revising the article; SNH, Analysis and interpretation of data, Drafting or revising the article; ZI, Conception and design, Drafting or revising the article; HB, Conception and design, Analysis and interpretation of data, Drafting or revising the article

## Author ORCIDs

Marc Deffains, http://orcid.org/0000-0003-0734-6541
Hagai Bergman, http://orcid.org/0000-0002-2402-6673

## Ethics

Animal experimentation: All experimental protocols were conducted in accordance with the National Institutes of Health Guide for the Care and Use of Laboratory Animals and with the Hebrew University guidelines for the use and care of laboratory animals in research, supervised by the institutional animal care and use committee of the faculty of medicine, the Hebrew University, Jerusalem, Israel (Ethical Application Reference Number: MD-15-14412-5 ). The Hebrew University is an Association for Assessment and Accreditation of Laboratory Animal Care (AAALAC) internationally accredited institute.

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
