## [Decision Letter]

Thank you for submitting your article "Subthalamic, not striatal, activity correlates with basal ganglia downstream activity in normal and parkinsonian monkeys" for consideration by *eLife*. Your article has been reviewed by three peer reviewers, and the evaluation has been overseen by a Reviewing Editor and a Senior Editor. The following individuals involved in review of your submission have agreed to reveal their identity: Robert Turner (Reviewer #2); Erwan Bezard (Reviewer #3).

The reviewers have discussed the reviews with one another and the Reviewing Editor has drafted this decision to help you prepare a revised submission.

The reviewers appreciated the importance and novelty of the study showing that STN is a major input driving function and dysfunction of the basal ganglia.

However, they ask for a few clarifications about the analyses of electrophysiology.

The reviewers also ask for detailed analyses of cell death in the MPTP model. So we would ask that if the authors have histological analyses of the animals used here that can permit conclusions about where the dopamine cell death mostly impacts, this should be added.

Reviewer #1:

The authors demonstrate that the STN is pivotal in driving basal ganglia output both in healthy non-human primates and following MPTP treatment to induce a parkinsonian condition. The result is clear and convincing and the methods rigorously conducted. All told this is very important work.

The finding does however beg two questions that I think should be discussed:

1) Why do presumed MSNs not lock to the striatal oscillations in the parkinsonian state? Is this simply a matter of firing rate or of how far these neurons are below threshold?

2) What then does the striatal outflow achieve if it is not patterned by input? Should we presume that what is demonstrated pertains to time and spatially averaged changes (over a few seconds and compiled from many neurons), and the striatal outflow may be more directly related to input for brief but functionally important periods and/or with a much more refined/focal patterning missed in the current analysis?

Related to the above, Figure 12 is interesting in showing that there are perhaps two types of LFP activity, even in the healthy state, albeit much more sparse in the latter; with and without β activity. This presence of β, even in health deserves highlighting more, but the authors might also elect to see if there is any MUA locking in striatum within the vicinity of the sub-population of LFPs with a β peak, and whether, in turn, these selected MUAs correlate with BG outflow activity at least in the Parkinsonian state. In short, with striatum being so much bigger than STN, with less convergence, do we have to look on a finer scale for input-output correlations?

Reviewer #2:

In this manuscript, Deffains et al. describe important new findings regarding the general organization of basal ganglia (BG) circuits. They report that the patterns of task-related activity observed in BG output nuclei resemble those of the STN more closely than those of spiny projection neurons in the striatum. Moreover, the abnormal neuronal activity associated with parkinsonism is relatively consistent across STN and output nuclei, but not present in striatal projection neurons. These results are taken as evidence that the STN is the primary driver of neuronal activity in BG output neurons under both normal and pathologic conditions.

The manuscript provides unambiguous results concerning an important clinically relevant topic. Appropriate methods are used for the most part and the results are well interpreted and integrated with existing literature. Added attention could be paid to a few aspects of the study.

Fraction of neurons modulated (Results): It would be helpful to see a stronger rationale for these results. Many readers will wonder why it is important to devote so many paragraphs of text and figures to the points made in this section. Do these results address a controversy? Do they provide ground breaking insights?

I/D balance in striatal neurons: It comes as little surprise that striatal projection neurons have a strongly positive I/D balance. Given their characteristic very low baseline firing rates, task related decreases in firing rate are nearly impossible to detect. I think it would be helpful to acknowledge this fact and explain how the similarity/dissimilarity analyses used go beyond a simple comparison of raw I/D balances.

Striatal neuron subtypes: STN and SNr populations show a striking change in I/D balance at cue offset. Is it possible that subtypes of MSN activity can account for the striking change in STN and SNr? It is virtually guaranteed that a variety of different response subtypes are included in the MSN population. Can the responsive MSNs be sorted into two general subgroups, one group that responds during cue presentation and another group that responds during the extended delay period? Inspection of Figure 4 suggests that is not the case, but it is worth addressing this point explicitly. Doing so would dispel one source of doubt that skeptics will bring to the manuscript.

Tests for similarity/dissimilarity: The manuscript does not include statistical comparisons of the similarity/dissimilarity of the activities of neuronal populations. This should be possible given the tremendous amount of detailed data collected. The Discussion describes the results as "STN activity correlated with […] the BG downstream network." Is it possible to do a correlation analysis?

Results in Discussion: Figure 17 is not mentioned until the Discussion. It is probably better to introduce these observations in the Results section.

STN output broadly distributed?: At several points, the Discussion describes STN output to be divergent or "broadly distributed." Evidence for such a divergence is slim. Mathai et al. (Mvt Disord 2013) addresses a different topic completely. Nambu (2002) provides no new evidence for such an organization.

Reviewer #3:

Deffains et al. aim at showing that subthalamic, not striatal MSN, activity correlates with basal ganglia downstream activity in normal and MPTP monkeys. This is an interesting study that regroups, in one experimental design, several notions that were so far scattered into different pieces of literature. In that respect it is a novel and informative study that takes advantage of well-defined methodologies routinely used in the contributing team. This reviewer wishes to express two concerns: the first is the lack of systematic approach caused by the use of a harsh MPTP intoxication procedure. While the first 6 figures deal with an elegant study of the whole BG physiology during a temporal discounting classical conditioning task, such investigation is not made possible by the "severe" parkinsonian syndrome displayed by the animals. Slower regimens of intoxication enable to produce dopa-responsive parkinsonian monkeys capable of operant behaviour. The resulting disconnection between part I (first 6 figures) and part II (following figures) sections dampen this reviewer's enthusiasm. The second concern is that no detailed anatomo-pathological study accompanies the comprehensive physiological investigations. It is surprising, especially given the low n of animals, that no post-mortem assessment has been performed. MPTP-induced dopamine neuron (and terminals) degeneration should have been studied so that the extent and pattern of neurodegeneration is characterized in the vicinity of recording tracks. A high magnification dopamine fibers (TH, DAT, VMAT) counting possibly associated with a direct anti-DA immunohistochemistry should be performed in all BG recorded structures in order to assess the striatal vs extra-striatal impact of MPTP-induced lesions. Indeed, MPTP induces a loss of DA terminals not only in the striatum but as well in all BG nuclei although with disparate intensities.

---

## [Author Response]

The reviewers appreciated the importance and novelty of the study showing that STN is a major input driving function and dysfunction of the basal ganglia.

However, they ask for a few clarifications about the analyses of electrophysiology.

We believe that we have clarified the referees' questions about the electrophysiology analyses in the revised manuscript. We have added several new analyses and figures as detailed in the responses below.

The reviewers also ask for detailed analyses of cell death in the MPTP model. So we would ask that if the authors have histological analyses of the animals used here that can permit conclusions about where the dopamine cell death mostly impacts, this should be added.

We thank the editors and the referees for this comment. We have a fluoro-dopa PET study of one of animals (Monkey K) and a histological analysis of the other animal (Monkey S, done in collaboration with Dr. Suzanne Haber, University of Rochester). Details are provided below in the responses to the referees' comments and in the revised manuscript.

*Reviewer #1:*

The authors demonstrate that the STN is pivotal in driving basal ganglia output both in healthy non-human primates and following MPTP treatment to induce a parkinsonian condition. The result is clear and convincing and the methods rigorously conducted. All told this is very important work.

We thank the referee for his summary and positive evaluation of our study.

The finding does however beg two questions that I think should be discussed:

1) Why do presumed MSNs not lock to the striatal oscillations in the parkinsonian state? Is this simply a matter of firing rate or of how far these neurons are below threshold?

We thank the referee for his comment and fully agree that the investigation on the incapacity of abnormal striatal local field potential (LFP) to entrain spiking activity of striatal projection neurons in parkinsonism should be expanded in both the frequency and temporal domains.

For this reason, we conducted additional analyses to determine whether or not the absence of locking between MSN spiking activity and abnormal oscillating (LFP) striatal synaptic inputs in the parkinsonian state is simply a consequence of the low firing rate of the MSNs or due to the high spike threshold of the MSNs. We believe that our new analysis supports the high spike-threshold hypothesis, i.e., the abnormal striatal synaptic inputs are not strong enough to depolarize the membrane potential of the MSNs above their threshold potential. To support the threshold hypothesis we carried out two new analyses:

First, we examined the coherence between the multi-unit activity (MUA) in the striatum (and elsewhere) and the LFP (New Figure 14—figure supplement 1) before and after MPTP intoxication. Unlike the low discharge rate of isolated MSNs (~2 Hz and below the frequency of the 8-15 Hz LFP oscillations; Figure 17 in the revised manuscript) the MUA recorded in the striatum in the vicinity of isolated MSN is representative of the discharge of many neurons; hence the coherence results are not confounded by the slow discharge rate of single MSN. Comparison of Figure 14—figure supplement 1 (MUA-LFP coherence) and Figure 14 (single-unit-LFP coherence) did not reveal any significant differences.

In addition, for each BG neuronal component, we calculated the spike-triggered averages of the LFPs (STA LFPs) recorded in the vicinity of the spiking neurons (i.e., from the same electrode; New Figure 16 in the revised manuscript). Again, this analysis overcomes the putative confounding effect of the low discharge rate of the MSNs, because if the MSN's spikes are triggered by the LFP oscillations but not at every cycle of them (Brunel and Hakim, Chaos. 2008; 18(2):015113; Kopell and LeMasson, PNAS. 1994; 91:10586-10590), one should see a locking between the spikes and the LFP oscillations. Indeed, although spikes of STN, GPe, SNr neurons and striatal TANs, were locked to the trough of the LFP oscillations after MPTP intoxication, striatal MSN spiking activity did not exhibit such locking.

Finally, to rule out the possibility that the STA LFP results in the parkinsonian state were influenced by the lower discharge rate of the MSNs, we randomly diluted the spike train of each neuron (MSNs included) firing at a higher rate than the mean discharge rate of the MSNs. After the random dilution, the discharge rate of all studied neuronal populations equaled the mean discharge rate of the MSNs and then we repeated the STA LFP analysis (New Figure 16, right column). The "diluted" STAs LFP resembled the original STAs, and thus supports our claim of a significant difference between striatal MSNs vs. STN and BG downstream neurons.

The description of the new methods and results are given in the text of the revised manuscript.

2) What then does the striatal outflow achieve if it is not patterned by input? Should we presume that what is demonstrated pertains to time and spatially averaged changes (over a few seconds and compiled from many neurons), and the striatal outflow may be more directly related to input for brief but functionally important periods and/or with a much more refined/focal patterning missed in the current analysis?

We thank the referee for his comment which has led to extensive changes in the Discussion. We believe that we now present a better view of the physiology and pathophysiology of the BG.

Briefly, we agree with the referee that in most brain areas the output (spikes) is shaped by the input (LFP). However, while we have no doubt that the output of the striatum (MSN spiking activity) is shaped by the striatal synaptic input, we would like to emphasize the possible differences between striatal LFP and striatal synaptic input.

First, there is still an on-going debate concerning the physical correlates of sub-cortical LFPs, especially when recorded by a mono-polar electrode (vs. a remote ground). If the striatal LFP mainly reflects cortical EEG, it might differ from the striatal synaptic input for several reasons: *(1)* only a fraction of the cortical neurons project to the striatum (Turner and DeLong, J. Neurosci. 2000; 20(18):7096-7108) and therefore cortical EEG might be different from the activity of cortico-striatal neurons; *(2)* the thalamus is a prominent glutamatergic input to the striatum (Smith et al., Trends Neurosci. 2004; 27(9):520-527) and thalamic activity, especially the activity of thalamic intra-laminar nuclei, might be hidden from the EEG point of view.

Even if striatal LFP reflects striatal sub-threshold (synaptic) activity, it probably reflects the (cortical and thalamic) glutamatergic activity and may only be marginally influenced by the lateral GABAergic connectivity or the dopaminergic input to the striatum. Modern striatal models concur that the dopaminergic input to the striatum plays a major role not only in modulation of the efficacy of the cortico-striatal synapses, but also as regards the excitability of the striatal projection neurons. Thus, coincidence between cortical and midbrain dopaminergic activity may be a necessary pre-condition for MSN spike generation.

In summary, we agree with the referee's suggestions that striatal outflow may be more directly related to input for brief but functionally important (probably marked by dopaminergic bursts) periods and with a much more refined pattern that was probably missed in the current population/average analysis. We have modified the Discussion of the revised manuscript to reflect these issues.

Related to the above, Figure 12 is interesting in showing that there are perhaps two types of LFP activity, even in the healthy state, albeit much more sparse in the latter; with and without β activity. This presence of β, even in health deserves highlighting more, but the authors might also elect to see if there is any MUA locking in striatum within the vicinity of the sub-population of LFPs with a β peak, and whether, in turn, these selected MUAs correlate with BG outflow activity at least in the Parkinsonian state. In short, with striatum being so much bigger than STN, with less convergence, do we have to look on a finer scale for input-output correlations?

We thank the referee for his insights and very valuable comment. We have made the necessary modifications in the revised manuscript to address this point.

First, we modified the text of the Discussion to highlight the presence of LFP β oscillations in the normal state and relate this finding to the extensive literature on the role of β oscillation in normal (status-quo, stop) behavior (Engel and Fries, Curr. Opin. Neurobiol. 2010; 20:156–165 and Brittain and Brown, Neuroimage. 2014; 85:637–647).

Second, following this comment, for each spike-LFP pair recorded in the parkinsonian state, we performed a linear regression analysis between the strength of the 8-15 Hz LFP oscillations and the degree of spike-field coherence. This analysis has been done using both single-unit activity (New Figure 15 in the revised manuscript) and MUA (New Figure 15—figure supplement 1) for the calculation of the spike-field coherence. Note that the linear regression analysis using MUA as representative of the spiking activity was done to be consistent with the MUA-LFP coherence analysis of the Figure 14—figure supplement 1). We preferred to conduct this continuous examination of the relationship between spiking activity and LFP oscillation because it does not impose a definition of an artificial threshold to determine whether or not LFP is oscillatory. Our results indicated that regardless of the BG neuronal component (i.e., including the striatal MSNs) the strength of the 8-15 Hz LFP oscillations was not related to the emergence of synchronization between spiking activity and LFP oscillations along the BG network after induction of parkinsonism. Thus, we can probably rule out the possibility that there was spike or MUA locking in striatum within the vicinity of the sub-population of LFP with a strong β peak.

Reviewer #2:

In this manuscript, Deffains et al. describe important new findings regarding the general organization of basal ganglia (BG) circuits. They report that the patterns of task-related activity observed in BG output nuclei resemble those of the STN more closely than those of spiny projection neurons in the striatum. Moreover, the abnormal neuronal activity associated with parkinsonism is relatively consistent across STN and output nuclei, but not present in striatal projection neurons. These results are taken as evidence that the STN is the primary driver of neuronal activity in BG output neurons under both normal and pathologic conditions.

The manuscript provides unambiguous results concerning an important clinically relevant topic. Appropriate methods are used for the most part and the results are well interpreted and integrated with existing literature. Added attention could be paid to a few aspects of the study.

We thank the referee for the comprehensive summary of our study and for highlighting the reliability of the results, as well as the rigor of the interpretations.

Fraction of neurons modulated (Results): It would be helpful to see a stronger rationale for these results. Many readers will wonder why it is important to devote so many paragraphs of text and figures to the points made in this section. Do these results address a controversy? Do they provide ground breaking insights?

In the revised manuscript we allocated more space to explaining the importance of the finding showing persistent fractions of modulated neurons along the BG main axis. Briefly, the BG main axis connects the brain areas encoding the current state of the subject and the brain areas controlling action. Therefore, it is important to show that persistent modulatory activity (i.e., during prolonged delays between sensory perception and action, especially when sensory information is no longer available) is not a unique property of the frontal cortex and other cortical areas, but can also be found along the BG main axis. Here, the demonstration that the BG main axis generates persistent modulatory activities is in line with the model that the BG main axis is able to convert transient sensory inputs into persistent internal representations. These BG persistent representations can be held and evaluated until they can be used later by the cortical and brainstem motor centers to guide the optimal action/response.

In addition, the evidence for persistent fractions of modulated neurons along the BG main axis, especially during the delay period is behind our finding that STN, not striatum, is correlated with the time-varying increase-decrease balance of spiking activity in BG downstream structures (Figure 6). Indeed, we found that the reversal of the increase/decrease discharge balance in the BG downstream structures during the delay period was concurrent with the relative changes in the increase/decrease discharge balance of the STN during the transition from cue to the delay period.

We modified the Discussion to better reflect the novelty and importance of these results concerning BG persistent activity.

I/D balance in striatal neurons: It comes as little surprise that striatal projection neurons have a strongly positive I/D balance. Given their characteristic very low baseline firing rates, task related decreases in firing rate are nearly impossible to detect. I think it would be helpful to acknowledge this fact and explain how the similarity/dissimilarity analyses used go beyond a simple comparison of raw I/D balances.

We agree with the referee's comment. Given the low discharge rate of striatal MSNs, it is unexpected to observe decreases in MSN discharge rate. Nevertheless, such decreases in the discharge rate of the MSNs (Figure 4) have been already described (e.g., Adler et al., J. Neurosci. 2012; 32(7):2473-2484; Samejima et al., Science. 2005; 310(5752):1337-1340; Báez-Mendoza et al., J. Neurophysiol. 2016; 115(1):68-79). We have made the necessary changes in the Discussion of the revised manuscript to clarify this point.

More importantly, in order to statistically compare the similarity/dissimilarity of the activities between BG input and downstream structures beyond a simple comparison of the raw I/D balance, we performed a cross-correlation analysis of the neuronal responses of the input structures (striatum or STN) and the BG downstream structures (GPe, GPi or SNr). Details of this new analysis are described in the Materials and methods of the revised manuscript in the section "Similarity/dissimilarity of the neuronal responses between BG input and downstream structures". Briefly, cross-correlations were calculated between the neuronal responses (relative PSTHs) of each BG input-downstream neuron pair. Each pair was clustered according to the (principal) polarity of the neuronal responses. This way, we obtained four distinct scenarios and examined the similarity/dissimilarity of the neuronal responses of neuron pairs exhibiting a similar (increase-increase or decrease-decrease) or opposite (increase-decrease or decrease-increase) polarity. We found that STN, compared to striatal MSN, responses to behavioral events were more strongly correlated with BG downstream responses. In the revised manuscript, these results are depicted in the new Figure 7 and incorporated into the main text of the Results section.

Striatal neuron subtypes: STN and SNr populations show a striking change in I/D balance at cue offset. Is it possible that subtypes of MSN activity can account for the striking change in STN and SNr? It is virtually guaranteed that a variety of different response subtypes are included in the MSN population. Can the responsive MSNs be sorted into two general subgroups, one group that responds during cue presentation and another group that responds during the extended delay period? Inspection of Figure 4 suggests that is not the case, but it is worth addressing this point explicitly. Doing so would dispel one source of doubt that skeptics will bring to the manuscript.

Like the referee, we also wondered whether the striking changes at cue offset in the increase/decrease discharge balance of BG downstream neuronal populations would account for two distinct subtypes of MSN activity. Examination of the temporal patterns of changes in activity in the MSN assembly during the delay condition (Figure 4) did not reveal two distinct sub-groups, such that one group responded during cue presentation and the other group responded during the delay period. Nevertheless, as the referee requested, we carried out a formal clustering analysis of the MSN responses. In Figure 18 we show the clustering analysis of the MSN activities during the immediate and delayed conditions. Briefly, for each trial condition, salient features of MSN activities (relative PSTHs) were extracted with principal component analysis, using the scores of the first two principal components (PCs) as features for clustering. Identification of different MSN activity clusters was accomplished via the k-means clustering method, using the silhouette method to determine the optimal number of clusters between the limit of 1 and 4. In both conditions, we obtained three distinct clusters (see Figure 18) reflecting three different MSN activity profiles (see Figure 18): unmodulated activity (black); increase in activity (red); decrease in activity (blue). Our clustering analysis revealed that MSN activity profiles represented two distinct MSN sub-populations that exhibited persistent increase or decrease in activity from cue onset to outcome delivery in both conditions. However, we found no evidence for two general MSN sub-groups, where one group responded during cue presentation and the other group responded during the extended delay period.

We added these observations to the Discussion of the revised manuscript. But we believe that there is no need to add this analysis to the main text.

Author response image 1.Clustering analysis of MSN activity profiles in the immediate and delayed conditions.**DOI:**
http://dx.doi.org/10.7554/eLife.16443.031

Tests for similarity/dissimilarity: The manuscript does not include statistical comparisons of the similarity/dissimilarity of the activities of neuronal populations. This should be possible given the tremendous amount of detailed data collected. The Discussion describes the results as "STN activity correlated with […] the BG downstream network." Is it possible to do a correlation analysis?

As discussed above, in the revised manuscript we provide details on a cross-correlation analysis of the activity of the input structures (striatum or STN) and the BG downstream structures (GPe, GPi or SNr; New Figure 7 and related text). As requested, this analysis quantifies the similarity/dissimilarity of the activities between BG input and downstream structures.

Results in Discussion: Figure 17 is not mentioned until the Discussion. It is probably better to introduce these observations in the Results section.

We agree with referee's suggestion. Figure 17 ("Firing rate in the BG network before and after MPTP intoxication") in the revised manuscript (i.e., Figure 13 in the previous submission) is now mentioned in the Results section.

STN output broadly distributed?: At several points, the Discussion describes STN output to be divergent or "broadly distributed." Evidence for such a divergence is slim. Mathai et al. (Mvt Disord 2013) addresses a different topic completely. Nambu (2002) provides no new evidence for such an organization.

Here, we believe that the reviewer is referring to Mathai and Smith (Front. Syst. Neurosci. 2011) and not Mathai et al. (Mov. Disorders. 2013). We did not quote the latter (Mathai et al., Mov. Disorders. 2013) whose main finding is that the primate STN is traversed by numerous myelinated axons which occupy as much as 36-45% of its associative and sensorimotor regions. However, this paper does not discuss the STN output to its pallidal and nigral targets.

Concerning the broad distribution of STN output and the review by Mathai and Smith (2011), referee is right; there is no clear evidence for broad STN distribution and we agree that this point is still being debated.

Our claim for an anatomical broad distribution of STN (vs. a more narrow distribution of the striatal output) is based mainly on the number of cells in the BG input and downstream structures. For example, Oorschot DE (J. Comp. Neurol. 1996; 366:580–599) found that the rat BG consists, on average, of 2.790,000 neostriatal neurons, 13,600 STN neurons, 46,000 GPe, 3,200 GPi, and 26,300 SNr neurons. Similar relationships were reported by Percheron et al. for non-human primates (The basal ganglia related system of primates: definition, description and informational analysis, The basal ganglia IV, 1994, pp 3-20) and by Yelnik for the human STN (Functional Anatomy of the Basal Ganglia, Mov. Disorders. 2002; 17(3):S15–S21).

In addition, single-axon tracing of STN neurons revealed a very broad distribution of their terminals in the GPe, GPi and SNr (Sato et al., J. Comp. Neurol. 2000; 424(1):142-152). Therefore, we believe that although quantitative data are still lacking, we can assume (in line with Mink, 1996; Nambu et al., 2002 and many others) that the STN output is broadly distributed in the BG downstream structures. Nevertheless, we made the necessary changes in the Discussion of the revised manuscript to clarify that evidence for STN output divergence is still slim.

*Reviewer #3:*

Deffains et al. aim at showing that subthalamic, not striatal MSN, activity correlates with basal ganglia downstream activity in normal and MPTP monkeys. This is an interesting study that regroups, in one experimental design, several notions that were so far scattered into different pieces of literature. In that respect it is a novel and informative study that takes advantage of well-defined methodologies routinely used in the contributing team.

We thank the referee for his comprehensive summary and support.

This reviewer wishes to express two concerns: the first is the lack of systematic approach caused by the use of a harsh MPTP intoxication procedure. While the first 6 figures deal with an elegant study of the whole BG physiology during a temporal discounting classical conditioning task, such investigation is not made possible by the "severe" parkinsonian syndrome displayed by the animals. Slower regimens of intoxication enable to produce dopa-responsive parkinsonian monkeys capable of operant behaviour. The resulting disconnection between part I (first 6 figures) and part II (following figures) sections dampen this reviewer's enthusiasm.

We agree with the referee that it would be very interesting to study the behavior and the BG neural correlates in animal models with slower regimens of MPTP intoxication (i.e., that produce dopa-responsive parkinsonian monkeys capable of operant behavior). The use of MPTP-treated monkeys that exhibit moderate parkinsonian symptoms is undoubtedly a necessary step that would broaden our understanding of the physiology and the pathophysiology of the BG, as well as the early stages of Parkinson's disease.

In the current study, we decided to use the more severe MPTP model that probably mimics end-stage Parkinson's disease (off dopaminergic medication) in order to capture the full spectrum of synchronous oscillations in the BG network. Indeed, previous studies (Leblois et al., Eur. J. Neurosci. 2007; 26(6):1701-1713) failed to find BG oscillatory activity in early stages of chronic MPTP treatment. In addition, it has been shown that severe MPTP-treated monkeys usually develop stable clinical symptoms (Potts et al., Exp. Neurol. 2014; 256:133-143). In our study, the stability of the clinical symptoms in the MPTP-treated monkeys was a critical issue. Given the relatively long period of recordings after induction of parkinsonism (due to the number of BG structures in which we had to record – striatum, STN, GPe and SNr) we had to guarantee the stability of the parkinsonian symptoms in both animals during the entire duration of the recordings to be able to further compare activities at the different stages of the BG.

We modified the text in the Discussion of the revised manuscript to better clarify these points.

The second concern is that no detailed anatomo-pathological study accompanies the comprehensive physiological investigations. It is surprising, especially given the low n of animals, that no post-mortem assessment has been performed. MPTP-induced dopamine neuron (and terminals) degeneration should have been studied so that the extent and pattern of neurodegeneration is characterized in the vicinity of recording tracks. A high magnification dopamine fibers (TH, DAT, VMAT) counting possibly associated with a direct anti-DA immunohistochemistry should be performed in all BG recorded structures in order to assess the striatal vs extra-striatal impact of MPTP-induced lesions. Indeed, MPTP induces a loss of DA terminals not only in the striatum but as well in all BG nuclei although with disparate intensities.

After completion of the neuronal recordings in the parkinsonian state, a dynamic fluoro-dopa positron emission tomography (PET) study and post-mortem tyrosine hydroxylase (TH) immunohistochemistry (Monkey K and S, respectively) were used to assess the severity of dopamine depletion following the MPTP treatment.

The details of the PET study and the post-mortem TH immunohistochemistry are given in the Materials and methods of the revised manuscript in the section "*Assessment of dopamine depletion in the MPTP-treated monkeys".* The results are incorporated into the main text and depicted in new Figure 8. Briefly, the results showed that both MPTP-treated monkeys (in comparison with healthy animals) exhibited a large dopamine depletion in the dorsal striatum (i.e., caudate nucleus and putamen) similar to that observed in the brain of human patients suffering from advanced Parkinson's disease. Therefore, at this stage, even though we cannot assess dopamine degeneration of the STN and BG downstream structures in the parkinsonian state, we are confident of the validity of our non-human primate model of Parkinson's disease. Nevertheless, we did the necessary changes in the Discussion of the revised manuscript to highlight this point.